# SABRE populates ER domains essential for cell plate maturation and cell expansion influencing cell and tissue patterning

**Xiaohang Cheng, Magdalena Bezanilla***

Department of Biological Sciences, Dartmouth College, Hanover, United States

**Abstract** SABRE, which is found throughout eukaryotes and was originally identified in plants, mediates cell expansion, division plane orientation, and planar polarity in plants. How and where SABRE mediates these processes remain open questions. We deleted *SABRE* in *Physcomitrium patens*, an excellent model for cell biology. *SABRE* null mutants were stunted, similar to phenotypes in seed plants. Additionally, polarized growing cells were delayed in cytokinesis, sometimes resulting in catastrophic failures. A functional SABRE fluorescent fusion protein localized to dynamic puncta on regions of the endoplasmic reticulum (ER) during interphase and at the cell plate during cell division. Without *SABRE*, cells accumulated ER aggregates and the ER abnormally buckled along the developing cell plate. Notably, callose deposition was delayed in Δ*sabre*, and in cells that failed to divide, abnormal callose accumulations formed at the cell plate. Our findings revealed a surprising and fundamental role for the ER in cell plate maturation.

## Introduction

Due to their sessile nature, plants cannot simply run away from environmental stimuli and instead must adjust to their environment by regulating growth patterns. Plant growth is a coupled process involving deposition of extracellular matrix material – the cell wall – around individual cells and cell expansion. Precise regulation of the composition of this matrix ensures where a particular cell can expand. To control cell shape throughout a tissue, polarity cues at the cellular and tissue level ensure coordination ultimately patterning whole organs, such as correctly oriented roots and stems (*Blilou et al., 2005*; *Kania et al., 2014*; *van Dop et al., 2020*), as well as specialized structures including stomata and root hairs (*Gilroy and Jones, 2000*; *Houbaert et al., 2018*; *Mansfield et al., 2018*; *Zhang et al., 2016*). Positioning the cell division plane contributes to cell shape and provides polarity information (*Zhang and Dong, 2018*). For example, in root and shoot tissue in seed plants, the cell division plane is perpendicular to the growth axis and creates polygonal cells that are aligned longitudinally with each other (*Ambrose et al., 2007*; *Galjart, 2005*; *Schaefer et al., 2017*; *Smith et al., 1996*; *Walker et al., 2007*), ensuring that expansion is aligned uniformly with the overall plant growth axis. For specialized cell types, polarized cell wall deposition and cell plate positioning cooperate to determine cell morphology. In the leaf epidermis, asymmetric cell divisions define the stomatal guard cells (*Dong et al., 2009*), and restricted expansion in cells defines the jigsaw-shaped epidermal cells (*Sapala et al., 2019*). In filamentous cells such as pollen tubes and root hairs in seed plants, polarized secretion of flexible wall material to the cell apex leads to cell expansion occurring only at the apex of the cell (*Bascom et al., 2018*; *Chen et al., 2018*; *Dehors et al., 2019*; *Orr et al., 2020*).

Using forward genetics, many studies have identified mutations that alter plant morphogenesis and have provided insights into the regulation of expansion and polarity, demonstrating that plants

**\*For correspondence:**
magdalena.bezanilla@dartmouth.edu

**Competing interests:** The authors declare that no competing interests exist.

regulate cell wall deposition in a myriad of ways. The *sabre* mutant, which has short fat roots, was first identified in Arabidopsis in the early 1990s (*Benfey et al., 1993*). The increased root diameter resulted from exaggerated radial expansion primarily in root cortex cells, suggesting that SABRE plays a role in regulating expansion of diffusely growing cells (*Aeschbacher et al., 1995*). A second copy of *SABRE*, named *KINKY POLLEN* (*KIP*), which is expressed most strongly in roots, pollen, and developing seeds, was identified in a screen for abnormal pollen tube and root hair morphology (*Procissi et al., 2003*). Plants lacking KIP form defective pollen tubes that exhibit periods of relatively normal growth interspersed with periods of slow or arrested growth. Recovery from these growth arrests often led to growth initiating in new directions, ultimately leading to the kinky or twisty phenotype. In addition, root hairs were shorter and thicker in *kip* mutants (*Procissi et al., 2003*). These data suggested that SABRE contributes to diffuse growth while KIP contributes to polarized growth. However, the homozygous *kip/sab* double mutant exhibited enhanced phenotypes in both diffuse and polarized growing tissues, indicating overlapping function of these two closely related genes (*Procissi et al., 2003*). More recent studies have found that in *sabre* mutants cell plate positioning in the root meristem was variable, resulting in cells that were not cylindrically aligned. Furthermore, root hair emergence was no longer restricted to the basal portion of the trichoblast cell (*Pietra et al., 2013*). In the *sabre* mutant, transcription factors that initiate root hair cell fate were also altered, resulting in the formation of root hairs from ectopic sites. Collectively these studies have pointed to a critical role for *SABRE* in regulating plant polarity at both cell and tissue levels (*Pietra et al., 2015*).

In *Zea mays*, *ABERRANT POLLEN TRANSMISSION 1* (*APT1*) gene was identified as the *SABRE/KIP* homolog, whose mutation also resulted in short and meandering pollen tubes (*Xu and Dooner, 2006*). Consistent with predictions of a Golgi localization sequence at the C-terminus of SABRE homologs (*Pietra et al., 2013*; *Xu and Dooner, 2006*), expressing C-terminal fragments of APT1 fused with fluorescent proteins in tobacco pollen tubes resulted in localization to the Golgi. However, full-length SABRE stably expressed in Arabidopsis exhibited punctate localization in the cytosol of root epidermal cells that did not obviously represent any known endomembrane compartment (*Pietra et al., 2013*). More detailed localization studies are needed to help reconcile these apparently contradictory findings.

Among plants, the moss *Physcomitrium* (formerly *Physcomitrella*) *patens* is an excellent cell biological model system and ideal for studying how cell shape affects developmental patterning (*Rensing et al., 2020*; *Rounds and Bezanilla, 2013*). Moss juvenile tissue, protonemata, is haploid and comprises a filamentous two-dimensional branching network that is a single-cell layer thick, making it readily amenable to high-resolution microscopy. Coupled with recent advances in CRISPR-Cas9-mediated genome editing allowing for rapid generation of null alleles and functional fluorescent fusion alleles (*Collonnier et al., 2017*; *Lopez-Obando et al., 2016*; *Mallett et al., 2019*), *P. patens* provides an opportunity to perform a detailed analysis of SABRE localization and function. Furthermore, in contrast to seed plants, *P. patens* has a single copy of *SABRE* and plants can be propagated asexually, avoiding potential problems resulting from defects in sexual reproduction.

Here, we generated a null *sabre* mutant and functional fluorescent fusions at the native genomic locus to investigate *SABRE* localization and function in *P. patens*. We found that Δ*sabre* plants are stunted, exhibiting defects in polarized growth, diffuse cell expansion, and dramatic cell division failures accompanied by deposition of brown material into the cytoplasm often resulting in cell death. Surprisingly, even with the polarized growth and division defects, SABRE did not localize to the cytoskeleton. Instead, SABRE localized to a fraction of the endoplasmic reticulum (ER) near the tip of the cell, at the cell cortex, and in the phragmoplast during cell division. These results indicate that SABRE regulates plant cell expansion and division via its interaction with the ER, pointing to a fundamentally important role for the ER in plant cell and tissue morphogenesis.

## Results

### Loss of SABRE function inhibits polarized growth and diffuse cell expansion

The *P. patens* genome has one *SABRE* gene that encodes for a 2736 aa protein. To disrupt *SABRE*, we used CRISPR-Cas9-mediated homology-directed repair (HDR) to insert a 363 bp cassette with

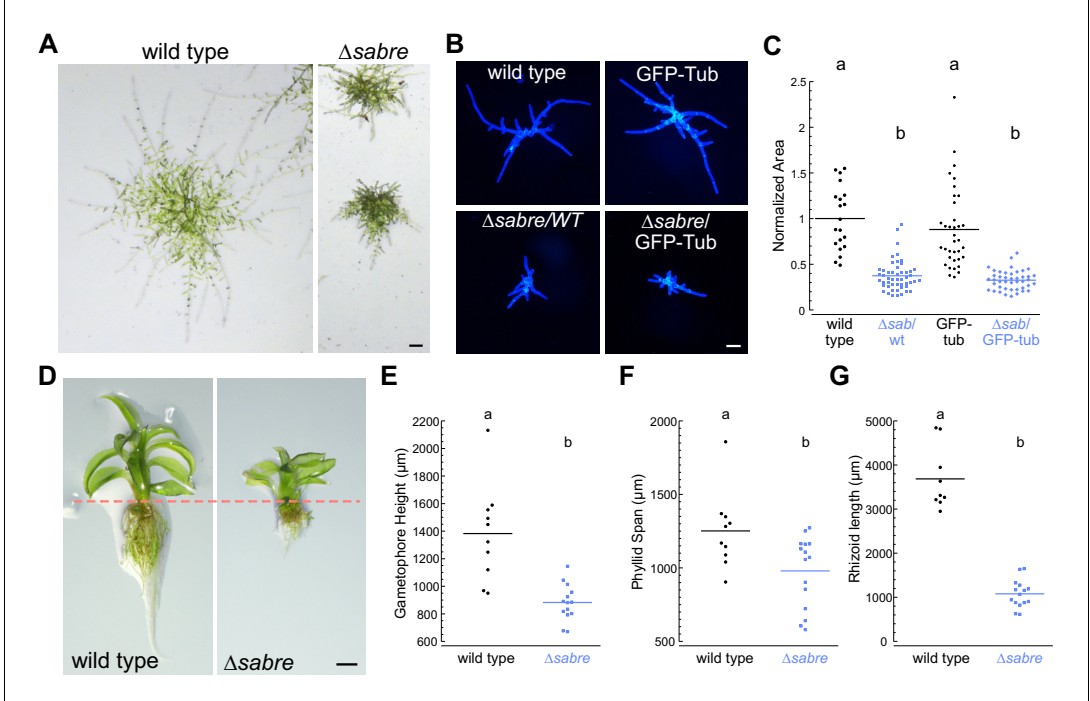

**Figure 1.** *SABRE* influences both polarized cell expansion and diffuse growth. (A) Example images of 2-week-old wild type and Δ*sabre* plants regenerated from single protoplasts. Extended-depth-of-focus (EDF) images were created from Z-stacks acquired with a stereomicroscope. Scale bar, 200 µm. (B) Representative fluorescence images of 7- day-old plants regenerated from protoplasts. Images of plants stained with calcofluor white were acquired with a fluorescent stereo microscope. Scale bar, 100 µm. (C) Quantification of Δ*sabre* plant size calculated from area of calcofluor fluorescence. Plant area was normalized to wild type. N = 20, wild type; N = 50, Δ*sabre*/WT; N = 36, GFP-tub; N = 43 Δ*sabre*/GFP-tub. Letters indicate groups with significantly different means as determined by ANOVA with a Tukey's HSD all-pair comparison post-hoc test (α = 0.05). For details of statistical analysis, see ***Supplementary file 1***. (D) EDF images of example mature gametophores. Scale bar, 500 µm. Dashed line indicates the boundary between the aerial tissue (top) and the rhizoids (bottom). (E–G) Quantification of gametophore height, phyllid span, and rhizoid length. Statistically significantly different means were determined by Student's *t*-test for unpaired data with equal variance, with p value indicated above the graphs. (E) N = 10, wild type; N = 14, Δ*sabre*. (F) N = 10, wild type; N = 15, Δ*sabre*. (G) N = 9, wild type; N = 15, Δ*sabre*.

The online version of this article includes the following source data and figure supplement(s) for figure 1:

**Source data 1.** Quantification of Δ*sabre* plant size calculated from area of calcofluor fluorescence.
**Source data 2.** Quantification of gametophore height, phyllid span and rhizoid length.
**Figure supplement 1.** Generating SABRE null mutant with stop cassette insertion.
**Figure supplement 2.** Early gametophore development.

stop codons in all three possible frames into exon 2 of the *SABRE* locus (***Figure 1—figure supplement 1***). The mutant allele results in a frame shift starting at amino acid 38 followed by a premature stop codon after amino acid 47. For consistency, all Δ*sabre* lines generated in this study contained the same mutation. To ensure that the mRNA expressed from the genome-edited allele was altered as predicted, we amplified the 5′ end of the *SABRE* cDNA isolated from wild type and Δ*sab* plants. We found that the cDNA from Δ*sabre* was larger due to insertion of the stop codon cassette (***Figure 1—figure supplement 1A***). Sequencing revealed that the transcript contained the expected in-frame stop codon.

*P. patens* protonemal tissue expands exclusively by polarized growth with cell division occurring in the apical cell of the filament. Subapical cells re-enter the cell cycle once a protrusion emerges generating a new filament, with the branching cell dividing at the base of the emerging protrusion. Compared to wild type, Δ*sabre* plants had smaller and more compact protonemata (***Figure 1A***). To quantify this difference, we regenerated plants from single protoplasts and measured the overall size 7 days after protoplasting (***Figure 1B***). Compared to protonemata in control plants, we found that Δ*sabre* protonemata were 60% smaller (***Figure 1C***). As protonemata age, some of the protrusions switch fates to bud-like structures that expand in three dimensions, ultimately resulting in the

development of leafy shoots, known as gametophores, the adult tissues. While protonemal filaments increase in size exclusively by polarized expansion at the tip of the apical cell, gametophores expand by diffuse growth (*Figure 1D*). We also observed that Δ*sabre* gametophores are 36% shorter than wild type (*Figure 1D, E*). The phyllids, leaf-like structures that emanate from the gametophore, were 22% smaller than wild type (*Figure 1F*). Rhizoids are polarized-growing filaments that grow from the base of the gametophore anchoring it in the soil. Δ*sabre* rhizoids were 70% shorter than wild type rhizoids, a decrease in size comparable to the polarized-growing protonemata (*Figure 1G*). Time-lapse imaging demonstrated that while early developmental patterning of Δ*sabre* gametophores was not altered, cell expansion was significantly delayed (*Figure 1—figure supplement 2*, *Video 1*). Together, these data demonstrate that the single *SABRE* gene regulates both polarized and diffuse growth in *P. patens*.

To determine whether smaller organ size resulted from changes in underlying cell size, we imaged protonemata and gametophore development at higher resolution (*Figure 2*). During protonemal development, the apical stem cell divides, leaving behind a subapical cell that does not elongate anymore. Thus, we measured the length of protonemal subapical cells. While Δ*sabre* cells were 30% shorter than wild type (*Figure 2A*), the decrease of overall plant size was 60% (*Figure 1C*). This discrepancy could be due to defects in the rate of growth or how often a particular filament is actively growing. To distinguish between these two possibilities, we measured cell expansion rates in actively growing apical cells as determined by time-lapse imaging. While on average actively growing Δ*sabre* cells grew only 20% slower than control cells, the reduction in growth was not statistically significant. In contrast, time-lapse imaging revealed that over the same time period Δ*sabre* protonemal filaments often grew significantly less than a comparable wild type filament due to the fact that for a large portion of the time-lapse acquisition the Δ*sabre* cell was not growing (*Figure 2B*, *Video 2*). As a result, we reasoned that additional mechanisms, such as frequent long pauses in growth, likely contributed to the decrease in plant size.

Interestingly, Δ*sabre* protonemal cells often changed direction during growth (*Figure 2B*, *Video 2*). Both actin and microtubules contribute to maintaining the direction of polarized growth. Cytoplasmic microtubules polymerize towards the cell tip where their plus ends then focus onto an apically localized actin spot (*Hiwatashi et al., 2014*; *Wu et al., 2018*). When actin filaments are disrupted, microtubules no longer focus below the tip (*Wu and Bezanilla, 2018*) and growth is inhibited. When microtubules are disrupted, the actin spot randomly appears and disappears throughout the cell, with expansion occurring in areas that accumulate actin, ultimately resulting in a loss of directional growth (*Wu and Bezanilla, 2018*; *Yamada and Goshima, 2018*). Since defects in growth directionality in Δ*sabre* cells resembled wild type cells lacking microtubules (*Doonan et al., 1988*; *Wu and Bezanilla, 2018*), we wondered if SABRE influenced the apical microtubule focus. We found that similar to wild type, the actin and microtubule foci in Δ*sabre* cells were persistently present at the tip (*Figure 2—figure supplement 1A, B*). Thus, the observed defects in growth directionality and rate appear to be independent of actin and microtubules. Actin and microtubules also form dynamic networks at the cell cortex in protonemal cells. Using variable angle epifluorescence microscopy (VAEM), we imaged microtubules and actin in wild type and Δ*sabre* plants and quantitatively compared the global dynamics of these networks (*Figure 2—figure supplement 1C, D*). We measured the correlation coefficient of the intensity of the GFP-tubulin and Lifeact-GFP (labeling the actin cytoskeleton) signal at all pixel locations for all temporal intervals (*Figure 2—figure supplement 1C, D*). As the microtubule and actin filaments change their position due to polymerization, depolymerization, and translocation, the correlation coefficient decreases with larger temporal increments (*Vidali et al., 2010*). A fast decay in the correlation coefficient is indicative of very rapid changes in global filament dynamics. Using this analysis, we did not observe obvious changes in microtubule organization or dynamics at the cortex (*Figure 2—figure*

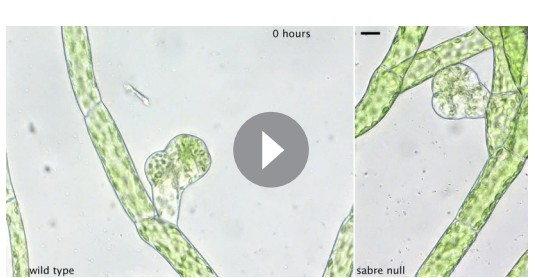

**Video 1.** Early gametophore development in wild type and Δ*sabre*. Each frame is an extended-depth-of-focus image generated from a brightfield Z-stack taken every 15 min. Scale bar, 10 µm. Video is playing at 10 fps.
https://elifesciences.org/articles/65166#video1

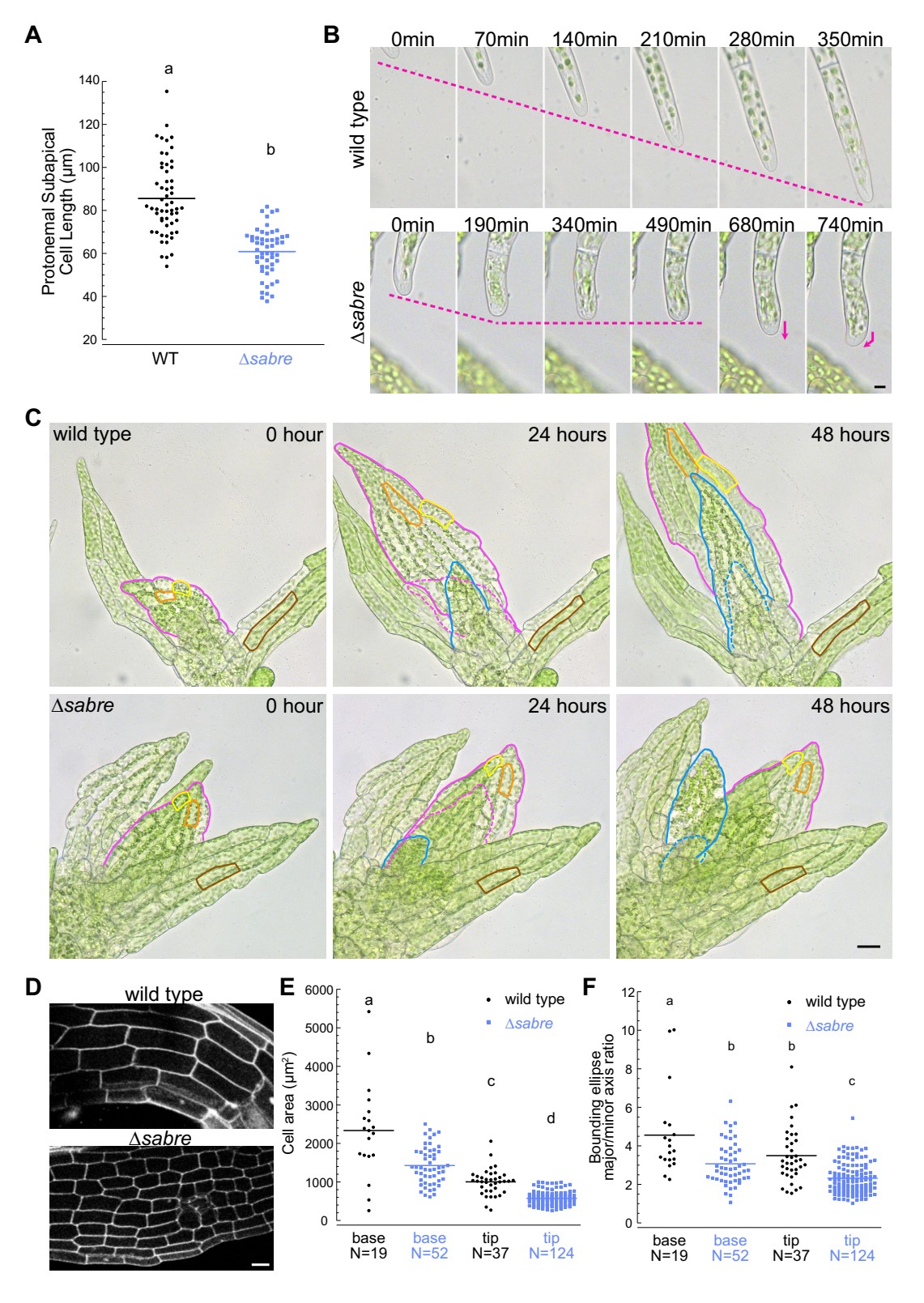

**Figure 2.** Reduced cell expansion underlies small plant size in Δ*sabre*. (**A**) Quantification of subapical cell length from 7-day-old plants regenerated from protoplasts. N = 56, wild type; N = 53, Δ*sabre*. Statistically significant different means were determined by Student's *t*-test for unpaired data with equal variance, with p value indicated above the graph. (**B**) Brightfield time-lapse images for wild type (top) and Δ*sabre* protonemata (bottom). Scale bar, 5 μm. Magenta dashed lines indicate the apical positions of the cells. Magenta arrows indicate growth directionality. Also see *Video 2*. (**C**)

*Figure 2 continued on next page*

*Figure 2 continued*

Brightfield time-lapse imaging of gametophores growing in microfluidic imaging devices. Magenta and blue lines indicate example phyllids 1 and 2, respectively, that expanded during the imaging period. Dashed lines in corresponding colors indicate the phyllid outlined 24 hr before. Orange and yellow lines outline example cells that expanded during the imaging period. Brown lines highlight example cells in a mature phyllid that did not obviously increase in size. Scale bar, 20 μm. Also see *Video 3*. (D) Example confocal fluorescent images of phyllids stained with propidium iodide used for quantification in (E, F). Scale bar, 30 μm. (E) Quantification of phyllid cell area. Base and top indicate cells were located at the base or top of the phyllid. (F) Quantification of the ratio of major/minor axis of the bounding ellipse fitted to each cell. Number of cells in each category indicated under the graph. Letters indicate groups that are significantly different as determined by one-way ANOVA with Tukey's HSD post-hoc test (α = 0.05). For statistical analysis details, see *Supplementary file 1* for (E) and 5 for (F).

The online version of this article includes the following source data and figure supplement(s) for figure 2:

**Source data 1.** Quantification of sub-apical cell length.
**Source data 2.** Quantification of phyllid cell area and shape.
**Figure supplement 1.** Organization and dynamics of actin and microtubule cytoskeletons in Δ*sabre*.
**Figure supplement 1—source data 1.** Quantification of cortical microtubule dynamics.
**Figure supplement 1—source data 2.** Quantification of cortical actin dynamics.

*supplement 1C*). However, cortical actin exhibited a slight increase in dynamics (*Figure 2—figure supplement 1D*).

Diffuse growing tissues in the *P. patens* Δ*sabre* mutants were stunted similar to what was observed in the Arabidopsis *sabre* mutant. Time-lapse imaging of expanding phyllids (*Figure 2*, *Video 3*) revealed that due to defective cell expansion (*Figure 2C*, yellow and orange lines), Δ*sabre* grew significantly less within the same time window compared to wild type (*Figure 2C*, magenta and blue lines and dashes). In mature phyllids, the final cell size was also smaller in Δ*sabre* (*Figure 2C*, brown lines). We quantified cell size in fully expanded phyllids and discovered that cell area at the base and tip of phyllids was reduced in Δ*sabre* (*Figure 2D–F*), consistent with the overall stunted gametophore stature (*Figure 1D–F*). In particular, cell area at the base of phyllids, which are generally the largest cells in the phyllid, was more affected than cells at the tip of the phyllid in Δ*sabre* (*Figure 2E*), suggesting that in the absence of SABRE cells may not be able to expand beyond a certain size. To measure changes in shapes of individual cells, we fit a bounding ellipse to each cell and quantified the ratio between the major and minor axes. Cells were less elongated in Δ*sabre* both at the tip and the base of the phyllids compared to wild type (*Figure 2F*).

## Loss of SABRE results in cytokinetic delays and can lead to failures in cytokinesis

During cytokinesis, plant cells use the phragmoplast, a microtubule-based structure, to build a new cell plate that physically separates the daughter cells. In protonemata, the phragmoplast forms

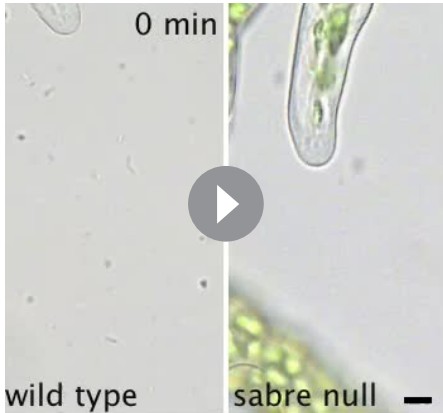

**Video 2.** *SABRE* influences polarized growth in protonemata. Brightfield images were taken every 10 min. Scale bar, 5 μm. Video is playing at 10 fps.
https://elifesciences.org/articles/65166#video2

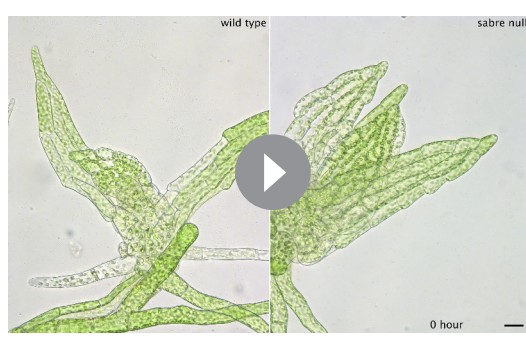

**Video 3.** *SABRE* influences diffused growth in gametophores. Extended-depth-of-focus images of brightfield Z-stacks taken every hour. Scale bar, 20 μm. Video is playing at 5 fps.
https://elifesciences.org/articles/65166#video3

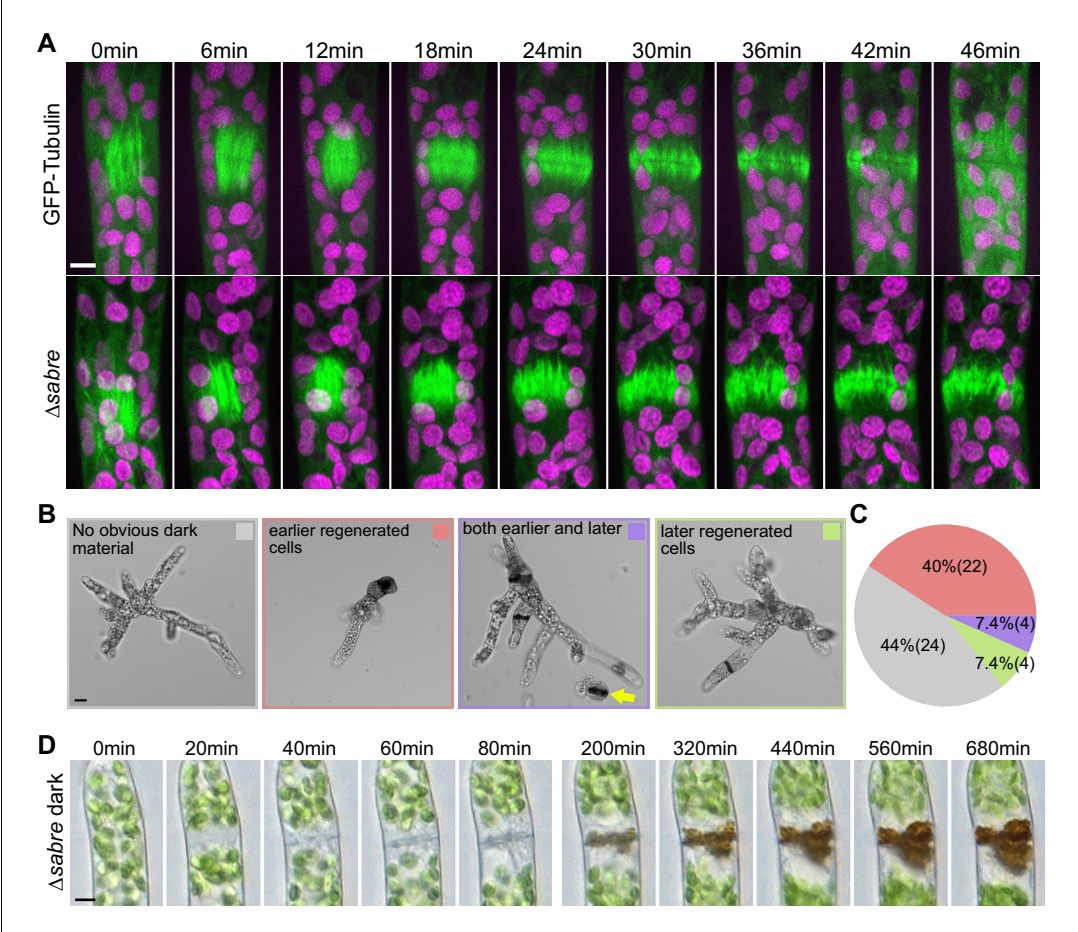

**Figure 3.** Delays in disassembling phragmoplast microtubules and quantification of cytokinesis failures during protonemal development. (A) Time-lapse imaging of phragmoplast microtubules. GFP-tubulin (green) and chlorophyll autofluorescence (magenta) are shown. First frames (0 min) occur within 2 min of nuclear envelope breakdown. Scale bar, 5 μm. Also see *Video 4*. (B) Representative images depicting brown material deposition near cell plates in 7-day-old plants regenerated from protoplasts. Colored frames correspond to categories quantified in (C). Yellow arrow indicates an example of a dead protoplast containing dark brown material at the first cell division site, resulting in the failure to regenerate. Scale bar, 20 μm. (C) Frequency of brown material deposits at different developmental stages (marked by cell shape and position in regenerated plant). Numbers in parentheses indicate numbers of plants. (D) Brightfield time-lapse extended-depth-of-focus images of brown material deposition in a cell that underwent cell division. Also see *Video 5*. Scale bar, 5 μm.

The online version of this article includes the following source data and figure supplement(s) for figure 3:

**Figure supplement 1.** Phragmoplast microtubule dynamics are not altered in Δ*sabre.*

**Figure supplement 1—source data 1.** Recovery rate of fluorescence for 1.5 minutes after the bleaching event.

between the two daughter nuclei, expanding perpendicular to the axis of the filament and eventually fusing with the plasma membrane. Phragmoplast microtubules are a dense bipolar array of microtubules that direct late secretory vesicles to the midzone where they fuse to form a membrane-encapsulated cell plate (*Smertenko et al., 2018*). During the early stage known as the disc phragmoplast, interdigitated microtubules with their plus ends at the cell equator are arranged in a spindle-like structure to establish the phragmoplast. As the cell plate grows, the microtubules label the outer edge of the expanding cell plate, known as the ring phragmoplast. Eventually the developing cell plate reaches the existing side wall and fuses with it, at which point the microtubule array dissipates (*Boruc and Van Damme, 2015*; *Smertenko, 2018*). In Δ*sabre* protonemata, we discovered that while mitosis is unaltered (e.g., *Figure 3A*, 0–12 min), phragmoplast microtubules were present for significantly longer than in control cells (*Figure 3A*, *Video 4*). We found that starting at the time of disc phragmoplast establishment (*Figure 3A*, 12 min), all control cell phragmoplast microtubules disappeared within 28–50 min (N = 10). However, for all Δ*sabre* cells (N = 10), microtubules were still

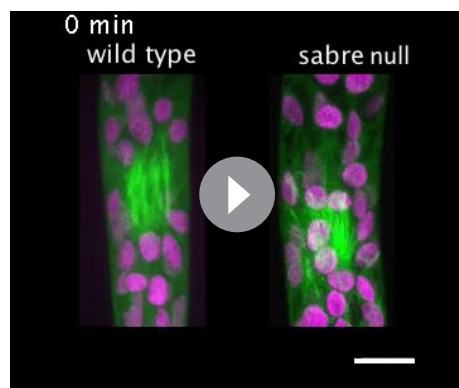

**Video 4.** Delays in disassembling phragmoplast microtubules in Δ*sabre* mutant. Images are maximum projections of confocal Z stacks of GFP-tubulin (green) and chlorophyll autofluorescence (magenta) in control and Δ*sabre* cells acquired every 2 min. Video is playing at 5 fps. Scale bar, 10 μm.
https://elifesciences.org/articles/65166#video4

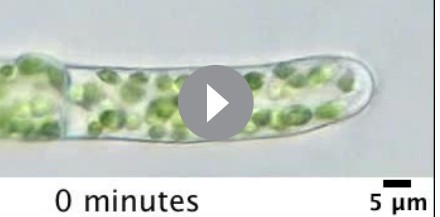

**Video 5.** Brown material deposition in Δ*sabre* occurs during cell plate formation. Each frame is a brightfield extended-depth-of-focus image taken every 10 min. Video is playing at 8 fps. Scale bar, 5 μm.
https://elifesciences.org/articles/65166#video5

present for at least 50 min after disc phragmoplast establishment. To test whether the delay in microtubule disassembly observed in Δ*sabre* phragmoplasts resulted from defects in microtubule dynamics, we performed photobleaching on phragmoplast microtubules labeled with GFP-tubulin (*Figure 3—figure supplement 1A*). Microtubule fluorescence recovered from photobleaching similarly in control and Δ*sabre* cells (*Figure 3—figure supplement 1B*), indicating that *SABRE* does not regulate microtubule dynamics during phragmoplast expansion.

Besides the lengthy delay in phragmoplast microtubule disassembly in protonemata, we observed that a fraction of cells contained dark brown material, which accumulated near the cell division plane (*Figure 3B–D*). We used numerous dyes for cell wall components to attempt to stain the brown material to get a hint of its composition, but none of these dyes stained. Time-lapse imaging revealed that brown material deposition was slow (*Figure 3D*, *Figure 3*, *Video 5*), but always initiated during a cell division event. Cells with brown material would either reinitiate growth after a long recovery time or die (*Figure 3*, *Video 5*). Cell death suggests that defects in cell plate formation resulted in loss of cell integrity. Interestingly, brown material appeared more frequently in cells with a large diameter, characteristic of the first few cells in plants regenerated from protoplasts (*Figure 3B, C*). Cells with a large diameter have inherently more degrees of freedom for orienting the phragmoplast. Furthermore, phragmoplast expansion and insertion occurs over a longer distance in these cells. We also noticed that Δ*sabre* protoplasts regenerated inefficiently compared to wild type, likely because many protoplasts died during the first cell division, with brown material deposited at the cell division site (*Figure 3B*, yellow arrow). Considering that neither cell length nor average growth rate accounted for the 60% reduction in protonemal area (*Figures 1* and *2*), we reasoned that the developmental delay caused by delays and failures in cytokinesis, coupled with frequent pauses in growth, likely accounts for the rest of the decrease in protonemal plant size.

## SABRE co-localized with a fraction of the ER at the cell cortex and in the phragmoplast midzone during cell plate maturation

To determine how *SABRE* impacts cell growth and division, we generated fluorescent fusions of *SABRE* to analyze its subcellular distribution. We used CRISPR-Cas9-mediated HDR to insert sequences encoding fluorescent proteins upstream of the stop codon of the native *SABRE* locus (*Figure 4—figure supplement 1A*). We demonstrated that tagging SABRE at its C terminus with either three tandem mEGFPs (*Vidali et al., 2009*) or mNeonGreens did not influence its function as we observed no growth defects in plants carrying only the tagged SABRE allele (*Figure 4—figure supplement 2A*). We focused on imaging the mNeonGreen fusion protein (hereafter, SAB-3mNG) because it was brighter than the GFP fusion. Confocal microscopy revealed that SAB-3mNG formed small puncta at the cell cortex, whose density was highest near the tip of the apical cell (*Figure 4A*). Even with mNeonGreen, the SAB-3mNG signal was weak and acquiring Z-stacks was challenging. To increase the signal, we inserted the maize ubiquitin promoter (a strong promoter) before the *SABRE-3mNG*

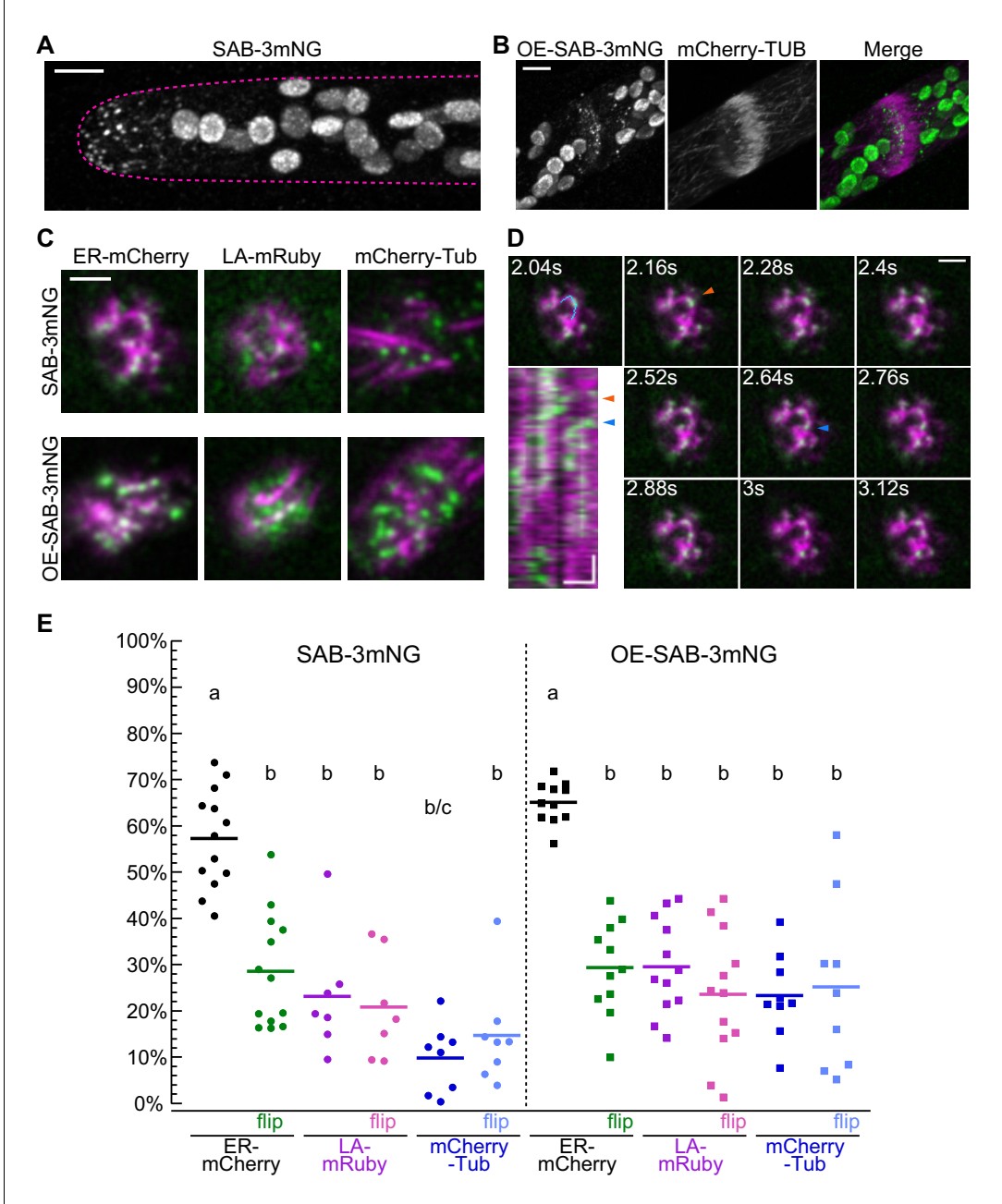

**Figure 4.** Localization of SABRE revealed by tagging SABRE at the C-terminus with three tandem mNeonGreen proteins. (**A**) SABRE-3mNeonGreen (SAB-3mNG) forms small puncta at the cell cortex that are more numerous near the tip of a growing apical cell. Image is a maximum projection of a deconvolved confocal Z-stack. Magenta dashed line indicates the outline of the cell. Scale bar, 5 µm. (**B**) Deconvolved confocal images of SABRE localization at the maturing cell plate. Scale bar, 5 µm. SAB-3mNG (green in merge) and mCherry-tubulin (magenta in merge) are shown. Large globular structures are chloroplasts, which autofluorescence in the mNeonGreen channel under these imaging conditions. (**C**) Variable angle epifluorescence microscopy (VAEM) images of SAB-3mNG (green) with mCherry-tubulin (mCherry-Tub), Lifeact-mRuby (LA-mRuby), and mCherry-KDEL (ER-mCherry) (magenta). Representative images are the first frame of a VAEM time-lapse acquisition. Scale bar, 2 µm. Also see *Video 6*. (**D**) VAEM time-lapse acquisition showing SAB-3mNG (green) moving along an endoplasmic reticulum (ER) tubule (magenta). Scale bar, 2 µm. Kymograph was generated using the cyan line drawn in the 2.04 s frame. Orange and blue arrowheads indicate two events when a SAB-3mNG puncta starts to move along the ER tubule. Horizontal scale bar, 1 µm. Vertical scale bar, 1.2 s. (**E**) Quantification of the fraction of SABRE area that overlapped with ER, actin, or microtubules for SAB-3mNG (left) and OE-SAB-3mNG (right). The area fraction is shown as a percentage for each category, with each data point representing the average of the first 50 frames of a time-lapse acquisition (120 ms interval) for one cell. For SAB-3mNG: N = 13, ER-mCherry; N = 7, LA-mRuby; N = 8, mCherry-Tub. For OE-SAB-3mNG: N = 11, ER-mCherry; N = 12, LA-mRuby; N = 9, mCherry-Tub. Letters indicate groups with

*Figure 4 continued on next page*

*Figure 4 continued*

significantly different means as determined by ANOVA with a Tukey's HSD all-pair comparison post-hoc test (α = 0.05). For statistical analysis details, see *Supplementary file 1*.

The online version of this article includes the following source data and figure supplement(s) for figure 4:

**Source data 1.** Quantification of the fraction of SABRE area that overlapped with ER, actin, or microtubules.
**Figure supplement 1.** Model for generating SABRE-3mNG tag and overexpression with maize ubiquitin promoter.
**Figure supplement 2.** C-terminal tagging and overexpression of SABRE does not influence protein function.
**Figure supplement 2—source data 1.** Quantification of plant area.
**Figure supplement 3.** SABRE forms dynamic puncta that were associated with the endoplasmic reticulum (ER).
**Figure supplement 3—source data 1.** Pearson's correlation coefficients of SABRE with ER, actin or microtubules.

start codon at the native genomic locus (*Figure 4—figure supplement 1B*). Plants with overexpressed SAB-3mNG (hereafter, OE-SAB-3mNG) had more SABRE puncta, but a similar localization pattern (*Figure 4—figure supplement 2C*). Disruption of this edited allele by inserting the stop codon cassette used to generate Δ*sabre* plants resulted in no fluorescence, indicating that Δ*sabre* alleles do not produce a protein (*Figure 4—figure supplement 2D*). Importantly, OE-SAB-3mNG plants were indistinguishable from SAB-3mNG plants, demonstrating that overexpression did not have any adverse consequences for protonemal growth (*Figure 4—figure supplement 2B*). During cell division, OE-SAB-3mNG formed discrete puncta decorating the entire developing cell plate (*Figure 4B*). Due to weak signals in both SABRE-3mNG and OE-SAB-3mNG, autofluorescence from the chloroplasts was prominently visible in the mNeonGreen channel for both lines.

During cell division, microtubules form the phragmoplast, actin interacts with microtubules and guides the expanding phragmoplast (*Buschmann and Müller, 2019*; *Müller, 2019*; *Wu and Bezanilla, 2014*), while the ER threads through the developing cell plate to build plasmodesmata – plant-specific channels that connect adjacent plant cells (*Sager and Lee, 2018*; *Tilney et al., 1991*). Since Δ*sabre* plants have severe defects in protonemal cell division, we generated SAB-3mNG and OE-SAB-3mNG in moss lines possessing microtubules (mCherry-tubulin), actin (Lifeact-mRuby), and ER (ER luminal marker SP-mCherry-KDEL) markers, enabling inquiry of SABRE behavior in the context of known phragmoplast structures. Initially, to maximize the SABRE signal, we used VAEM to image SABRE simultaneously with these markers at the cell cortex. Consistent with the finding that microtubules were not affected in Δ*sabre* plants (*Figure 2—figure supplement 1A, C*) and that SABRE localizes to the nascent cell plate even in late phragmoplasts that lack microtubules (*Figure 4B*), we found that cortical SABRE puncta did not associate with microtubules (*Figure 4C*, *Figure 4*, *Video 6*). We observed limited overlap between SABRE and actin at the cell cortex (*Figure 4C*, *Figure 4*, *Video 6*). However, most surprisingly, we discovered that cortical SABRE dynamically associated with ER tubules (*Figure 4C*, *Figure 4—figure supplement 3A, B,*, *Video 6*). In a kymograph along one tubule highlighted by the blue line, there were two events where SAB-3mNG dots translocated along the tubule (*Figure 4E*, arrowheads). More examples are shown in *Figure 4—figure supplement 3A, B*.

To quantify the degree of overlap between SABRE, actin, and microtubules, we used NIS-Elements software to calculate the Pearson's correlation coefficient between SABRE and the ER, actin, or microtubules. Pearson's correlation coefficients were highest between ER and SABRE (*Figure 4—figure supplement 3D*). Since the Pearson's correlation coefficient measures the correlation between signal intensities, these measurements were sensitive to overexpression and all correlation coefficients uniformly increased when SABRE was overexpressed (*Figure 4—figure supplement 3E*). To quantify co-

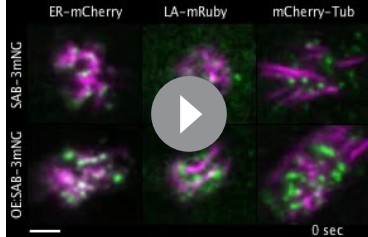

**Video 6.** Representative variable angle epifluorescence microscopy (VAEM) time-lapse acquisitions showing localization of SABRE with microtubules, actin, or endoplasmic reticulum (ER). SAB-3mNG or OE-SAB-3mNG (green), and ER (mCherry-KDEL), microtubules (mCherry-tubulin) or actin (LA-mRuby) (magenta) are shown. Time lapse acquired every 120 ms. Video is playing at 10 fps. Scale bar, 2 μm.
https://elifesciences.org/articles/65166#video6

localization independent of signal intensity, we measured the SABRE fraction that did not overlap with the ER, actin, or microtubules in the first 50 frames of a time-lapse acquisition (*Figure 4—figure supplement 3C*). We derived the overlapping SABRE fraction by subtracting the non-overlapping fraction from 1. Using this method, we found that on average 57% of SABRE puncta area overlapped with cortical ER tubules (*Figure 4E*). To determine if this overlap fraction occurs by chance, we measured the overlap using a flipped ER image (*Figure 4—figure supplement 3C*). The average overlap fraction plummeted to 29% when the ER image was flipped. Notably, 23% was the overlap measured with actin and was independent of whether the actin image was flipped (*Figure 4E*), suggesting that overlap with actin is coincidental. There was even less overlap with microtubules (*Figure 4E*), which increased to the same level as the overlap with actin when the microtubule image was flipped. Overexpression of SABRE increases the number of SABRE puncta at the cortex. However, importantly overexpression did not significantly affect the ER overlap (*Figure 4E*). With a higher density, the overlap with actin and microtubules increased to 25–30% and was the same even with flipped actin and microtubule images, suggesting that this degree of overlap occurs by chance.

Since Δ*sabre* exhibited serious defects in cell plate maturation and associated with the ER at the cell cortex, we further investigated the timing of SABRE and ER localization during cell division. After mitosis, the disc phragmoplast was labeled with thin strands of ER parallel to the microtubules and very little ER was present in the phragmoplast midzone (*Figure 5—figure supplement 1A*, first time point). In the ring phragmoplast, microtubules were shorter and expanded with the phragmoplast edge (*Figure 5—figure supplement 1*), while the ER strands parallel to the microtubules became more defined and the ER accumulated in the midzone perpendicular to the microtubules lining the future cell plate (*Figure 5—figure supplement 1A*). Cell plate maturation occurs during the late phragmoplast stage (*Smertenko et al., 2017*), coincident with an increase in the ER signal (third and later time points of *Figure 5A*, B, *Figure 5—figure supplement 1*). Confocal time-lapse imaging revealed that OE-SAB-3mNG localized to the midzone of the ring phragmoplast weakly, and the signal strengthened as the phragmoplast fully expanded and inserted into the side wall (*Figure 5A, B*, *Figure 5—figure supplement 1B*). The strongest OE-SAB-3mNG signal correlated with the timing of cell plate maturation (*Figure 5A, B*, *Video 7*) and remained until the ER signal visibly split in two on either side of the new cell plate (last two time points of *Figure 5A*).

## Loss of SABRE impacts the ER during interphase and at the cell plate during cell division

Given the striking association between SABRE puncta and the ER tubules at the cell cortex, we wondered if loss of SABRE function might impact ER in tip-growing cells and during cell division. With an ER luminal marker, SP-GFP-KDEL, we examined the overall ER structure in the cytoplasm. We discovered that Δ*sabre* cells contained abnormal ER aggregates in the cytoplasm in protonemata, which might underlie defects in polarized growth and directionality (*Figure 6A*, *Video 8*). Initially during cell division ER localization was normal (0 min time point of *Figure 6B*, *Video 9*). However, during the transition to an increase in ER parallel to the cell plate, the cell plate ER signal noticeably buckled in Δ*sabre*, while in control cells the ER was straight (*Figure 6B*, magenta arrowheads, *Video 9*). Interestingly buckling in Δ*sabre* mutants occurred coincident with the timing of maximal SABRE accumulation on the cell plate in control cells (*Figure 5*). Note that while twisting of the ER was obvious in the center of the cell plate, the edges adjacent to the side wall remained fixed, indicating that phragmoplast guidance mechanisms and the phragmoplast insertion site were not affected (*Figure 6B*, *Figure 6*, *Video 9*). For Δ*sabre* cells that divided relatively normally, the twisted ER, which was observed in all Δ*sabre* cells (N = 31), eventually straightened out 20–30 min later (*Figure 6B*, *Figure 6*, *Video 9*).

In addition to ER buckling, we noticed that nuclei in Δ*sabre* apical cells exhibited aberrant motility during cell division. With the ER outlining the re-formed nuclear envelope after mitosis, we observed that in control cells the apical daughter nucleus (*Figure 6—figure supplement 1A*) moved apically (phase I, blue arrows, see also *Figure 6C*), basally (phase II, orange arrows, see also *Figure 6C*), and then resumed apical migration (phase III, white arrows, see also *Figure 6C*) as has been described previously (*Yamada and Goshima, 2018*). In Δ*sabre* cells, the initial apical movement of the nucleus was delayed in 39% of the 31 imaged cells (*Figure 6—figure supplement 1B*, green arrowheads, *Video 10*), resulting in a close association between the nucleus and the cell plate, and coinciding with ER buckling (*Figure 6B*, *Figure 6—figure supplement 1B*, magenta arrowheads, *Video 10*).

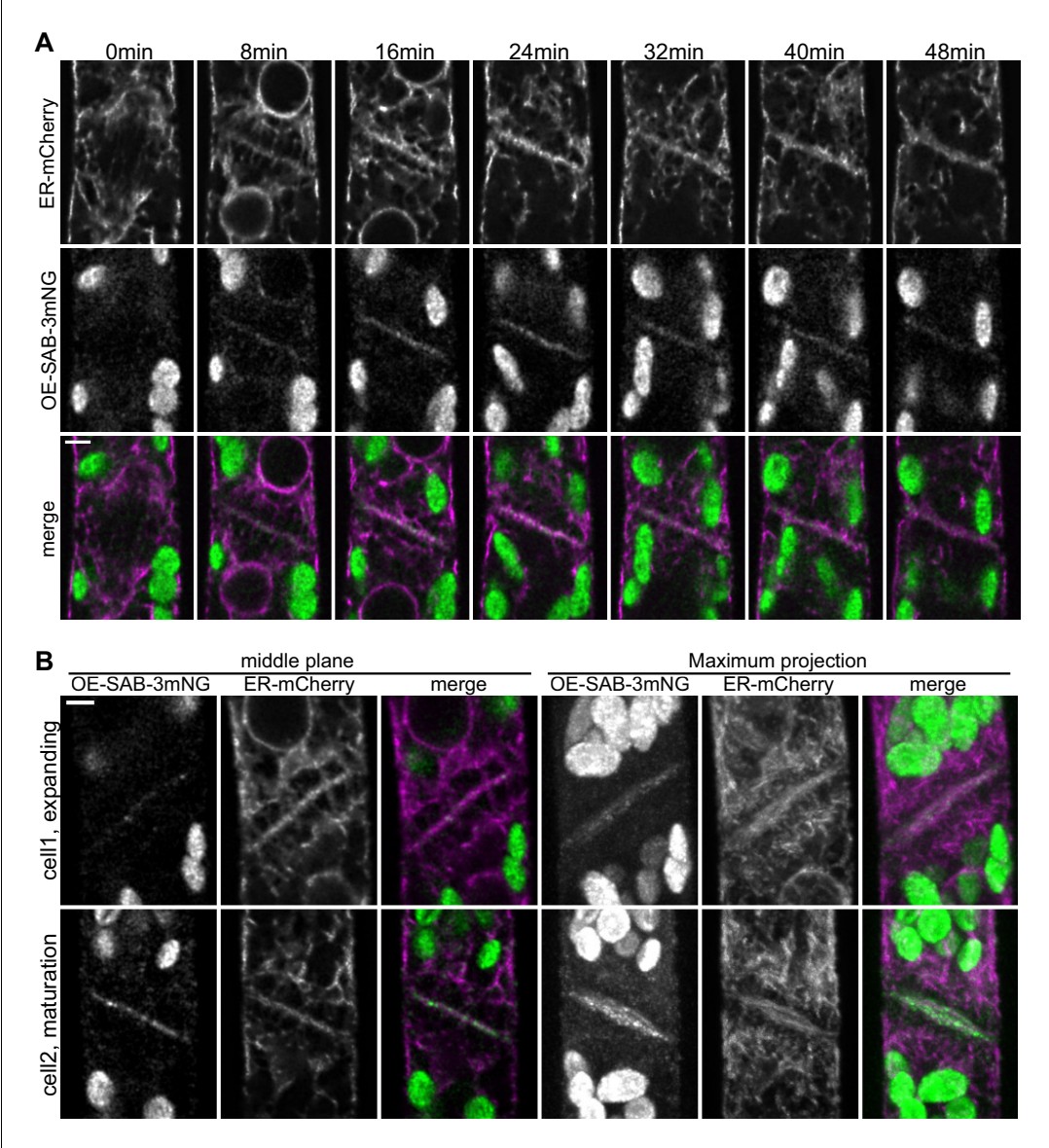

**Figure 5.** SABRE and endoplasmic reticulum (ER) accumulate on the nascent cell plate during cell plate maturation. (**A**) Deconvolved confocal images of OE-SAB-3mNG (green in merge) and mCherry-tubulin (magenta in merge) during cell division. Individual frames are medial focal planes of the cell. Scale bar, 3 μm. Also see *Video 7*. (**B**) Two representative cells with ER-mCherry (magenta in merge) and OE-SAB-3mNG (green in merge) showing the difference in the accumulation of SABRE and ER during cell plate expansion and maturation. Scale bar, 3 μm.

The online version of this article includes the following figure supplement(s) for figure 5:

**Figure supplement 1.** Timing of endoplasmic reticulum (ER), microtubule, and SABRE localization during cell plate formation.

While the phase I apical nuclear movement was normal in the remaining 61% of imaged Δ*sabre* cells, the subsequent basal movement was dramatically exaggerated in all cells (*Figure 6—figure supplement 1*, red arrowheads, *Video 10*).

To further quantify nuclear migration defects, we disrupted *SABRE* in a line expressing a nuclear localized GFP (NLS-GFP-GUS) and a plasma membrane marker (SNAP-TM-mCherry) (*van Gisbergen et al., 2018*), enabling imaging of cell division and nuclear movement before and after cell division (*Figure 6C*, *Video 11*). In control cells, basal migration (phase II) was often subtle or sometimes even appeared to be a stationary phase (*Figure 6C*; *Yamada and Goshima, 2018*). In contrast, Δ*sabre* basal (phase II) nuclear movement was extreme (*Figure 6C*, *Video 11*). In many cases, the nucleus migrated so far back that it appeared to deform as it smashed up against the new cell plate

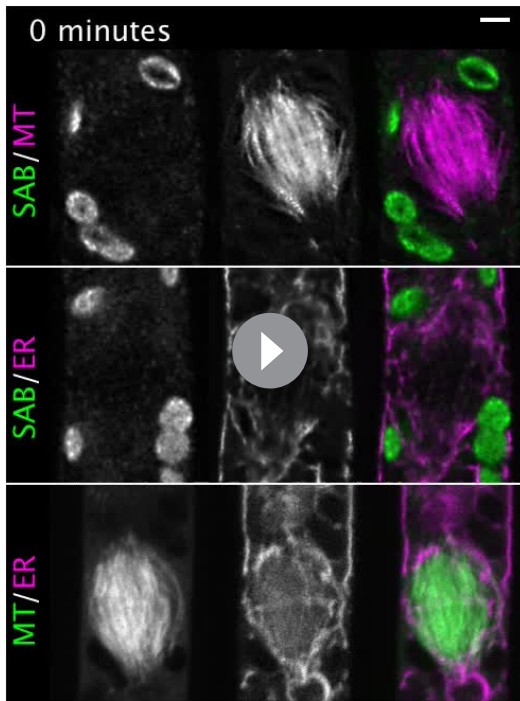

**Video 7.** SABRE accumulation correlates with endoplasmic reticulum (ER) localization at the phragmoplast and both signals accumulated during later stages of cell division. Images are single focal planes in the medial section of the cell. OE-SAB-3mNB (SAB), mCherry/GFP-tubulin (MT), and ER-mCherry (ER). Frame interval is 1 min. Video is playing at 5 fps. Scale bar, 3 μm.

https://elifesciences.org/articles/65166#video7

(*Figure 6C*, *Video 11*). To quantify the defect in basal movement, we measured the distance between the nucleus and the cell plate when the nucleus was closest to the cell plate (basal position), normalizing the basal position to the apical cell length at that time point. Despite shorter cells in Δ*sabre*, the relative basal nuclear position was significantly smaller than in control cells (*Figure 6D*). Beyond the immediate nuclear migration defects after cytokinesis, we also observed less consistent velocity and directionality during interphase in Δ*sabre*, represented by the zigzagging trajectory in the kymograph (*Figure 6C*, yellow arrows, *Video 11*). We tracked the nucleus and quantified the ratio between the final nuclear displacement and the total distance traveled in interphase. We discovered that Δ*sabre* nuclear movement was less linear and the ratio was smaller (*Figure 6E*) compared to wild type. The average instantaneous velocity of the nucleus in Δ*sabre* was expectedly smaller since the cells were not always actively growing (*Figure 6F*). Given that SABRE influences the ER and that the nuclear envelope is contiguous with the ER, defects in nuclear migration likely result from altered ER function.

To characterize key components of the cell division machinery in Δ*sabre* cells that failed to form a normal cell plate and resulted in cell death, we imaged the ER, Lifeact-GFP labeling actin and GFP-tubulin labeling microtubules. We observed the ER signal transition from defined tubules (*Figure 7A*, 40 min) in the phragmoplast to a diffuse signal (*Figure 7A*, 50 min), correlat-

ing with the onset of division failure. As the brown material accumulated, it was outlined by the ER signal (*Figure 7A*, *Figure 7*, *Video 12*). Both Lifeact-GFP and GFP-tubulin signals were relatively normal up until the phragmoplast had expanded to the parental cell membrane. However, at that point Lifeact-GFP became uneven around the phragmoplast with large gaps appearing near the presumptive cell plate. Lifeact-GFP lingered near the cell plate as the brown material generated a gap in the fluorescence gradually 'invading' the cytoplasm, which was accompanied by flashes of Lifeact-GFP fluorescence (*Figure 7B*, 50 min and after). In young plants regenerating from protoplasts, phragmoplasts labeled with GFP-tubulin expand across a larger distance, often finishing insertion on one side of the cell and then extending to the other (*Figure 7C*). In Δ*sabre* cells that accumulated brown material, phragmoplast microtubules on the outer edge became disoriented once they reached the parental plasma membrane (*Figure 7C*, 160 min). Later, deposition of brown material formed a gap between the remaining phragmoplast microtubules (*Figure 7C*). Sometimes cell plate defects resulted in immediate death. These cells lysed quickly after division and exhibited similar trends to cells that accumulated brown material; phragmoplast expansion was normal until it reached the side wall, at which point the cell lysed (*Figure 7—figure supplement 1*). Taken together our data suggest that phragmoplast expansion is normal in Δ*sabre*. However, during cell plate maturation, defects arise in Δ*sabre* cells; the ER becomes diffuse and both the actin and microtubule cytoskeletons remain associated with the cell plate.

## SABRE impacts callose deposition

To follow membrane and cell wall remodeling events occurring during cell division, we imaged cell division in the presence of the lipophilic dye, FM4-64, and the callose-specific dye, aniline blue.

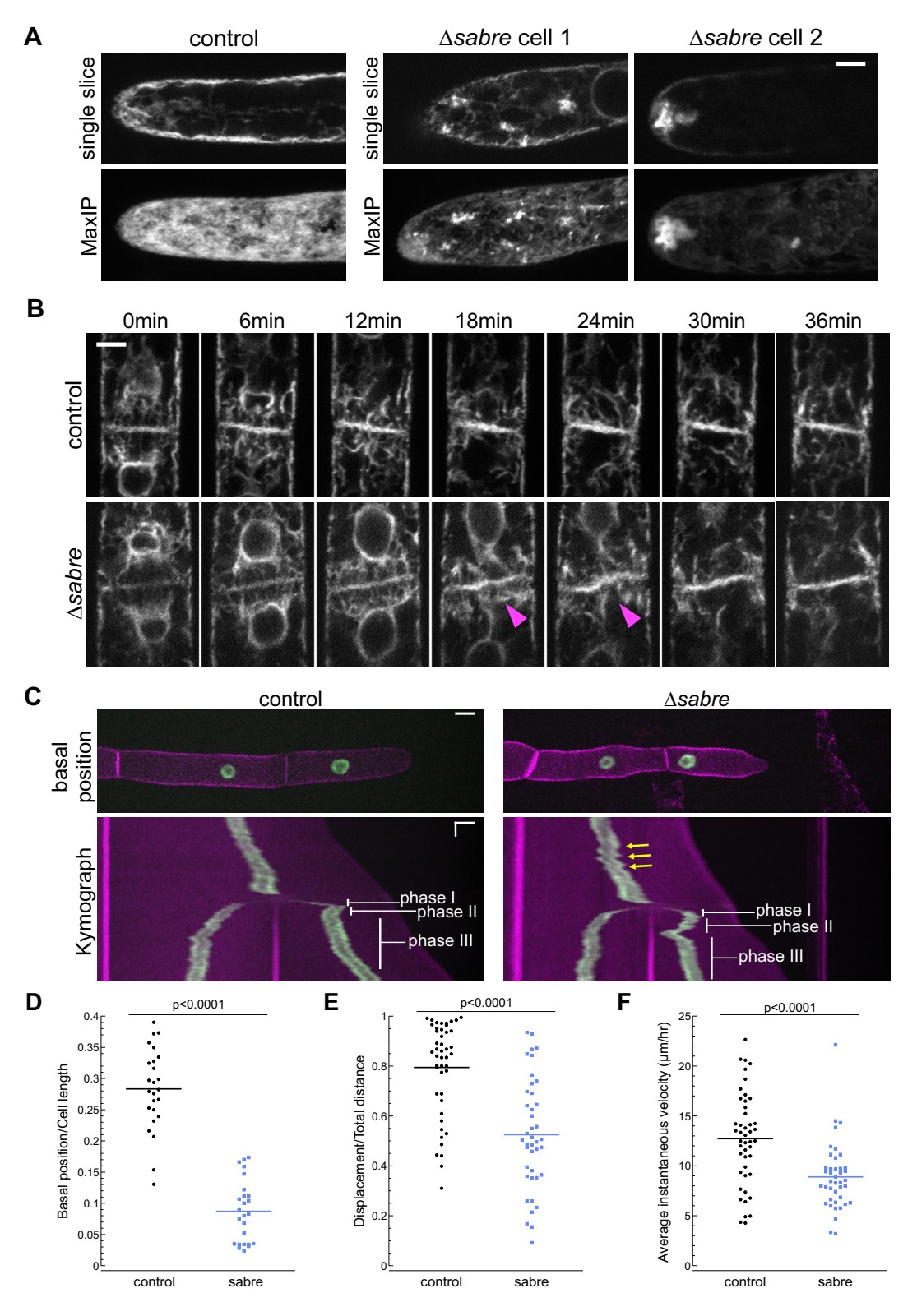

**Figure 6.** SABRE influences the endoplasmic reticulum (ER) and nuclear migration. (**A**) Example images from one control cell and two Δ*sabre* cells expressing SP-GFP-KDEL, which localizes to the ER lumen. Top: single focal plane. Bottom: maximum projection of confocal Z-stacks. Scale bar, 5 μm. Also see *Video 8*. (**B**) ER during cell division in wild type and Δ*sabre* cells. First frame is within 3 min of nuclear envelop reformation. ER buckles at the nascent cell plate in Δ*sabre* (magenta arrowheads). Scale bar, 5 μm. Also see *Video 9*. (**C**) Top panel: representative images of control and Δ*sabre* cells

*Figure 6 continued on next page*

*Figure 6 continued*

when the nucleus in the tip cell is closest to the new cell plate (basal position). NLS-GFP-GUS (green) accumulates in the nucleus, and SNAP-TM-mCherry (magenta) labels the plasma membrane. Scale bar, 10 μm. Bottom panel: kymograph created by drawing a line in the middle of the cell along the growth axis. Horizontal scale bar, 10 μm. Vertical scale bar, 1 hr. (D) Distance of the nucleus in the apical cell to the newly formed cell plate when it is at the basal position normalized to cell length. N = 25. (E) Quantification of linearity of nuclear movement during apical moving phases (I and III). Linearity determined by the ratio of total distance traveled divided by total linear displacement. Nuclear movement was tracked with TrackMate plugin in Fiji generating distance and displacement of nucleus. N = 46, control; N = 41, Δ*sabre*. (F) Quantification of nuclear velocity for the same cells and migration phases in (E). Average instantaneous velocity (μm/hour) was determined by dividing the total distance traveled by the time. Statistical analyses for (D–F) were performed using Student's *t*-test for unpaired data with equal variance, with p value indicated above the graphs. Also see *Video 11*.

The online version of this article includes the following source data and figure supplement(s) for figure 6:

**Source data 1.** Quantification of nuclear movement.

**Figure supplement 1.** SABRE influences the endoplasmic reticulum (ER), impacting nuclear movement and ER morphology at the nascent cell plate.

FM4-64 labels endocytic membranes (*Anieno and Robinson, 2005*; *Jelínková et al., 2010*; *Klima and Foissner, 2008*; *Kutsuna and Hasezawa, 2002*; *Tse et al., 2004*; *Ueda et al., 2004*; *van Gisbergen et al., 2008*), which are readily incorporated into the membrane surrounding the nascent cell plate early in cytokinesis as clathrin-mediated endocytosis is integral for remodeling the tubular membrane network during early phragmoplast formation (*Dhonukshe et al., 2006*; *Lam et al., 2008*; *Zhang et al., 2011*). However, once the phragmoplast has expanded across the entire mother cell, FM4-64 labeling decreases, consistent with a transition to different membrane trafficking machinery employed specifically during cell plate maturation and membrane fusion (*Drakakaki, 2015*; *Park et al., 2014*; *Smertenko et al., 2017*). In seed plants, callose is deposited early accumulating during the ring stage of the phragmoplast and reaching a peak just prior to fusion of the membrane encapsulating the nascent cell plate with the parental plasma membrane (*Samuels et al., 1995*). By imaging actively dividing wild type cells, we observed that aniline blue fluorescence steadily increased following a characteristic decrease in FM4-64 staining (*Figure 8A*, *Figure 8*, *Video 13*). Notably, aniline blue only stained cell plates that had fully expanded, suggesting that during phragmoplast expansion, callose within the membranous tubular network is not accessible to aniline blue in the extracellular environment. However, as FM4-64 levels diminished, which occurred during cell plate maturation and fusion of the cell plate membrane with the parental plasma membrane, aniline blue could access the callose, leading to the observed increase in staining (*Figure 8A*). Δ*sabre* cells that divided relatively normally (contained no brown material) exhibited a similar decrease in FM4-64 intensity (*Figure 8B*, solid lines). However, the FM4-64 signal continued to decrease for an additional 10 min, before aniline blue began to rise. At this time, wild type cells would have been fully stained with aniline blue. Further, the rate of increase in aniline blue was significantly slower in Δ*sabre* cells (*Figure 4B*, dashed lines), suggesting that it either took longer for aniline blue to diffuse into the cell plate or that callose was deposited later in Δ*sabre* cells.

To distinguish between defects in membrane fusion or delays in callose deposition, we analyzed images of cell division in protonemata labeled with a nuclear GFP and the membrane marker SNAP-TM-mCherry (*Figure 6C*). In these lines, loss of GFP fluorescence from the nucleus signaled nuclear envelope break down and the

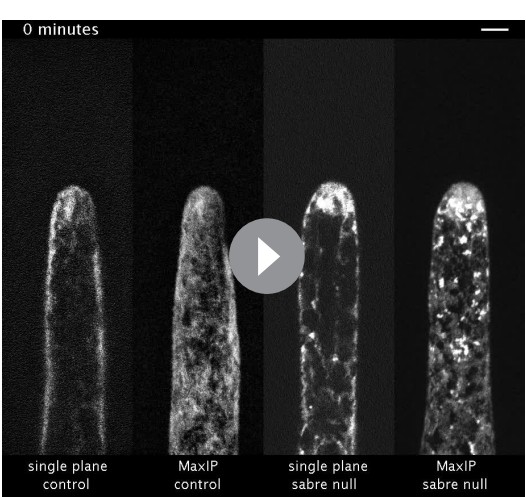

**Video 8.** Δ*sabre* exhibits bright endoplasmic reticulum aggregates in the cytoplasm. Images are single focal planes in the medial section and maximum projections of confocal Z-stacks of the cells showing SP-GFP-KDEL in control and Δ*sabre*. Confocal Z-stacks acquired every 10 min. Video is playing at 4 fps. Scale bar, 5 μm.
https://elifesciences.org/articles/65166#video8

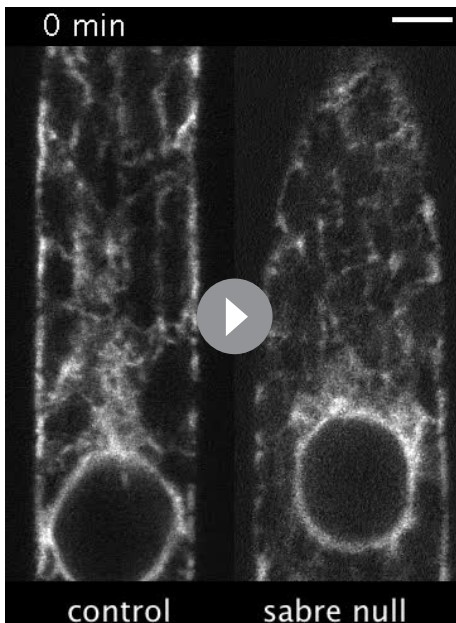

**Video 9.** Endoplasmic reticulum buckles during cell plate formation in Δ*sabre*. Each frame is a single focal plane in the medial section of the cell, taken every 3 min. Magenta arrow indicates buckling. Video is playing at 5 fps. Scale bar, 5 μm.

https://elifesciences.org/articles/65166#video9

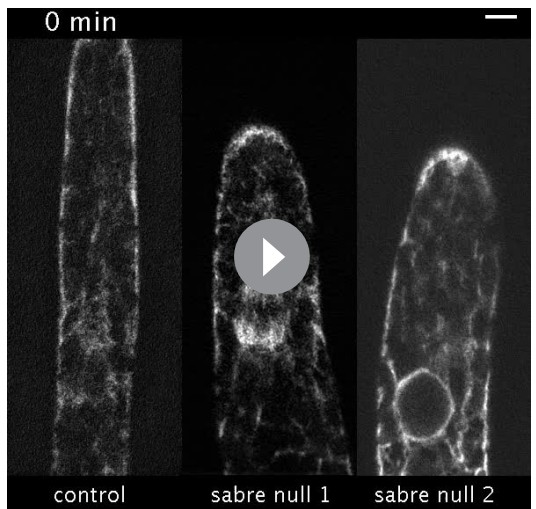

**Video 10.** Time-lapse imaging showing exaggerated nuclear basal migration in relation to phragmoplast endoplasmic reticulum (ER) buckling during cell plate formation. Δ*sabre* cell 1 showed delayed apical nuclear movement in phase I and exaggerated basal movement in phase II. Δ*sabre* cell 2 showed only exaggerated basal movement in phase II. Both cells exhibited ER buckling at the cell plate. Images are single confocal images taken every 10 min. Video is playing at 4 fps. Scale bar, 5 μm.

https://elifesciences.org/articles/65166#video10

onset of prometaphase. In contrast to FM4-64, SNAP-TM-mCherry did not accumulate on the nascent cell plate (*Figure 6*, *Figure 8C*, *Video 11*). Instead, SNAP-TM-mCherry appeared at the cell plate 30 min after nuclear envelope break down, the timing of which coincides with the end of phragmoplast expansion (*Figure 3A*). At the same time, SNAP-TM-mCherry disappeared from the plasma membrane adjacent to the cell plate (*Figure 8C*, white arrowheads), suggesting that SNAP-TM-mCherry accumulates at the cell plate by diffusing into the cell plate membrane from the parental plasma membrane once membrane fusion has occurred, rather than by delivery to the cell plate via exocytosis. By measuring the intensity of SNAP-TM-mCherry at the plasma membrane adjacent to the cell plate and in the middle of the cell plate, we quantified the kinetics of SNAP-TM-mCherry as it disappeared from the plasma membrane and appeared in the cell plate. Within 20 min after nuclear envelope break down, SNAP-TM-mCherry fluorescence decreased at the plasma membrane accompanied by an increase in the cell plate SNAP-TM-mCherry fluorescence. Interestingly Δ*sabre* cells exhibited the same SNAP-TM-mCherry kinetics as wild type, suggesting that diffusion from the parental plasma membrane was not impaired and thus membrane fusion is unaffected in Δ*sabre*.

To determine why cells that accumulated brown material often lysed, we used aniline blue staining to image accessible callose in cells with brown material. We found that compared to normal cell plates, which exhibited donut-shaped callose enrichments (*Figure 8E*), cells with brown material could be grouped into three categories: 27% stained weakly with aniline blue, 20% had a partial aniline blue ring, and 53% had large chunks of aniline blue staining near the brown material (*Figure 8F*; N = 15 cells). Time-lapse imaging in a Δ*sabre* cell that accumulated dark material and abnormal chunks of callose revealed accumulation of callose near the stalled microtubules that subsequently disappeared and reaccumulated on the other side of the cell (*Figure 8G*, *Video 14*), suggesting that Δ*sabre* cells exhibit defects in callose secretion and remodeling of callose during cell plate maturation.

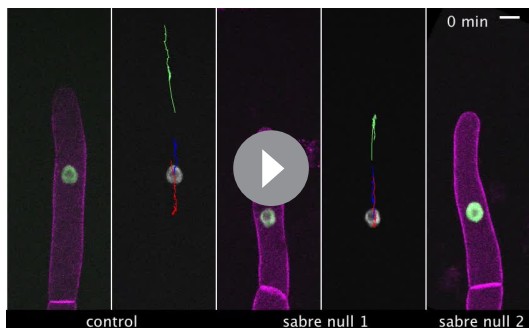

**Video 11.** Nuclear movement during tip growth and cell division. Blue traces represent the mother nuclei, and red and green traces represent the daughter nuclei in the apical and subapical cells, respectively. Nuclei are shown in green or gray with colored tracks, and plasma membrane shown in magenta. The column on the far right depicts a Δ*sabre* cell showing impaired forward nuclear movement before cell division and exaggerated basal nuclear movement after cell division. The nucleus moved basally all the way to the newly formed cell plate, appearing to smash into the cell plate, with an evident change in nuclear shape. The nucleus flattens out when in contact with the cell plate, observed at 510–525 min. Video is a maximum projection of confocal Z-stacks taken every 5 min. Video is playing at 8 fps. Scale bar, 10 μm.
https://elifesciences.org/articles/65166#video11

## Discussion

The *SABRE* gene was identified nearly three decades ago. In the intervening time, SABRE has been found to influence cell growth and polarity in plants. Despite the rich phenotypic analyses of mutants in seed plants, understanding the subcellular structures that *SABRE* localizes to and functions through has remained elusive. Here, we used a combination of genetics and live-cell imaging to study *SABRE* in the model bryophyte *P. patens*. Similar to *sabre* phenotypes in roots and shoots in Arabidopsis, Δ*sabre* plants in moss were stunted as a result of defects in cell expansion. Both polarized-growing protonemata and diffusely expanding cells in the phyllids of gametophores were smaller in Δ*sabre* plants. Similar to defects in pollen tubes from mutants of *KIP*, a second *SABRE* gene in Arabidopsis (**Xu and Dooner, 2006**), we observed twisty protonemata that also had periods of normal growth interspersed with pauses. However, unlike pollen tubes or root hairs, moss protonemata also undergo cell division, which was dramatically affected in Δ*sabre* plants. In extreme cases, defects during cell division led to failure characterized by loss of cell integrity and subsequent cell death. Interestingly, failures in cell division were readily detected in protonemata, but not in gametophores, suggesting possible tissue-specific SABRE functions during cell division.

Current models of cell plate formation posit that Golgi- and trans-Golgi-derived vesicles accumulate at the phragmoplast midzone where they fuse to each other to form a tubular network whose lumen accumulates callose (**Smertenko et al., 2017**). As the phragmoplast expands, more vesicles are added to the edge and the tubular network enlarges. During expansion, clathrin-mediated endocytosis remodels the membrane transforming the tubular network into a fenestrated sheet, which coincides with the peak of callose accumulation (**Samuels et al., 1995**). Ultimately this sheet fuses with the parental cell membrane (**Boruc and Van Damme, 2015**; **de Keijzer et al., 2017**; **Smertenko, 2018**; **Smertenko et al., 2017**). Once fusion occurs, the lumen of the cell plate becomes continuous with the apoplast, the extracellular environment of the plant, and the cell plate matures into a primary wall accompanied by changes in carbohydrate composition (**Drakakaki, 2015**). The exact mechanisms that dictate carbohydrate maturation are unclear, but throughout cell plate formation numerous components of vesicle trafficking and endocytosis have been shown to influence cell plate formation with distinct spatiotemporal contributions (**Chow et al., 2008**; **Lauber et al., 1997**; **Rybak et al., 2014**; **Smertenko et al., 2017**; **Steiner et al., 2016**). Microtubules and actin in the phragmoplast are hypothesized to direct vesicle trafficking as well as influencing cell plate positioning and structural stabilization of the nascent cell plate.

Our data demonstrate that SABRE plays a role in cell plate maturation influencing the timing of callose deposition. For Δ*sabre* cells that accumulated brown material, we observed a range of different callose staining behavior and time-lapse imaging revealed that large callose aggregates were not static, suggesting that cells lacking SABRE exhibit unregulated callose deposition and remodeling. In seed plants, callose begins to accumulate during the ring phragmoplast stage (**Park et al., 2014**; **Samuels et al., 1995**), coincident with the timing of SABRE and ER recruitment along the cell plate (**Figure 5**). EM studies have shown that the ER accumulates parallel to either side of the developing cell plate once the fenestrated sheet has fully expanded (**Seguí-Simarro et al., 2004**). However, whether ER in this region simply repopulates a new cortical ER domain or whether it plays a

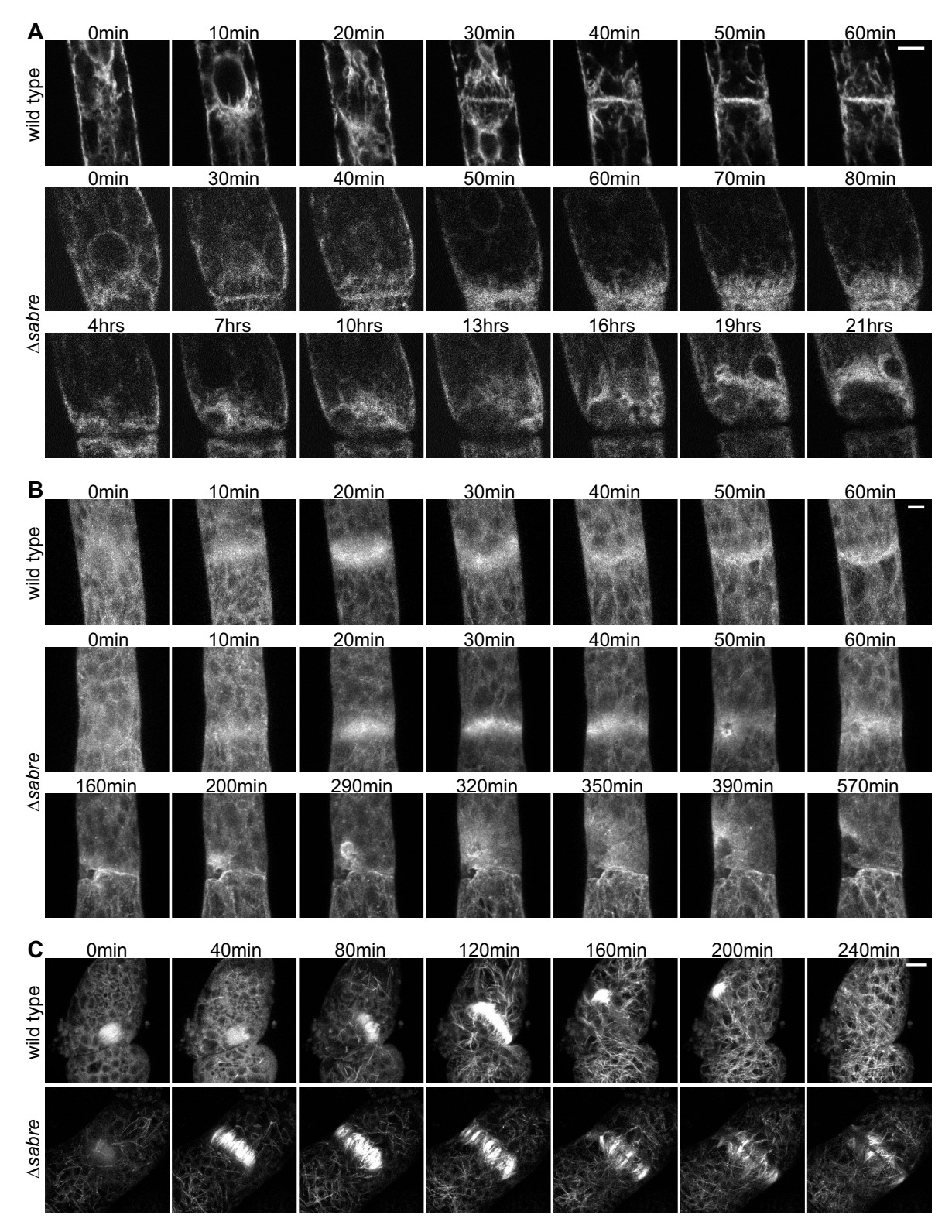

**Figure 7.** Behavior of endoplasmic reticulum (ER), actin, and microtubules during cytokinesis failures in Δ*sabre*. Single focal planes from confocal images of (**A**) ER labeled with SP-GFP-KDEL, maximum projection of confocal Z-stacks of (**B**) actin labeled with Lifeact-GFP, and (**C**) microtubules labeled with GFP-tubulin in wild type and Δ*sabre* cells that accumulate brown material during cell division. (**A**) Between 0 and 20 min, a phragmoplast

*Figure 7 continued on next page*

*Figure 7 continued*

formed. At 30 min, the Δ*sabre* phragmoplast failed to mature and brown material deposition gradually occurred over the following hours. Scale bar, 6 µm. (B) Scale bar, 5 µm. (C) Scale bar, 10 µm. Also see *Video 12*.

The online version of this article includes the following figure supplement(s) for figure 7:

**Figure supplement 1.** Phragmoplast microtubule and actin behavior during cytokinesis failures in Δ*sabre* that resulted in cell lysis during cell division with no brown material deposition.

role in cell plate maturation has been unclear. Here, we provide striking evidence that the ER domains decorated with SABRE at the nascent cell plate play a critical role during cell plate maturation. Without SABRE, the ER still accumulates, but in a fully expanded phragmoplast, the ER parallel to the cell plate invariably buckles in the middle of the cell. In Δ*sabre* cells, we observed delayed/unregulated callose deposition. Perhaps to lend structural support to the nascent cell plate lacking uniform callose, both actin and microtubules were retained at the expanded phragmoplast. Taken together our data suggest that the SABRE-decorated ER domains contribute to regulating callose deposition. Whether this regulation is direct or via secretion of callose synthase proteins to the nascent cell plate remains to be determined.

Beyond cell division, we found that SABRE influences the directionality and persistence of polarized growth and nuclear migration by altering ER function, not the cytoskeleton. Using a combination of confocal microscopy and VAEM, we discovered that SABRE co-localizes to regions of the ER and does not impact cytoskeleton localization or dynamics. In comparison to confocal, the increased signal-to-noise ratio afforded by VAEM revealed that at the cortex SABRE puncta associated with and moved along ER tubules. These results differ from previous work in Arabidopsis that had indicated *SABRE* influences microtubule cortical organization and preprophase band positioning and does not localize to the ER, Golgi, or TGN (*Pietra et al., 2013*). In contrast to Arabidopsis cells, protonemata do not have an organized cortical microtubule array or a microtubule-based preprophase band, suggesting that these differences could result from cell type variability. Furthermore, without the benefit of increased sensitivity enabled by VAEM, it would be very difficult to identify SABRE association with just a fraction of the ER, raising the possibility that SABRE may also associate with the ER subdomains in Arabidopsis.

We discovered that Δ*sabre* cells exhibited exaggerated basally directed nuclear movement during and after cell division. Nuclei in the apical cell also oscillated backwards as they moved apically towards the cell tip in Δ*sabre* cells. Microtubule motor proteins that mediate these nuclear movements during cell division have been identified in *P. patens*. Kinesin-14 drives basal movements (*Yamada and Goshima, 2018*), and kinesin-13 drives apical nuclear movement during prophase (*Leong et al., 2020*). Mutations in these motor proteins resulted in exaggerated movements in the opposite direction rather than a stationary nucleus, indicating that nuclear movement results from a balance of forces. The exaggerated basal nuclear migration in Δ*sabre* could either be the enhancement of basal moving forces or inhibition of the apical moving force, possibly generated by the ER or ER-localized proteins.

How the ER might influence polarized growth persistence via *SABRE* is an interesting question. The ER accumulates just below the cell tip where both actin and microtubules drive and steer polarized growth, respectively in protonemata. However, in Δ*sabre*, both the actin and microtubule cytoskeletons were not affected, suggesting that SABRE's impact on cell expansion is independent of the cytoskeleton. Of note, a recent study demonstrated that protonemata with impaired COPII function, which mediates ER to Golgi transport, exhibited aggregated ER and polarized growth defects (*Chang et al., 2020*),

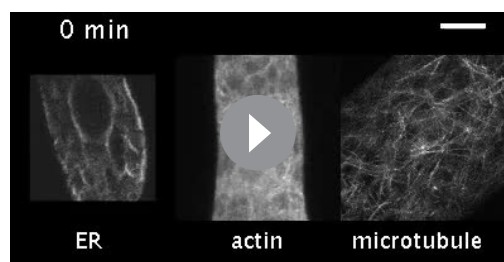

**Video 12.** Endoplasmic reticulum (ER), actin, and microtubule behavior during brown material deposition and cell division failure in Δ*sabre*. Single focal plane of ER (SP-GFP-KDEL), maximum projection of actin (Lifeact-GFP), and microtubules (GFP-tubulin) are shown. Time-lapse imaging was acquired every 10 min. Video is playing at 5 fps. Scale bar, 10 µm.

https://elifesciences.org/articles/65166#video12

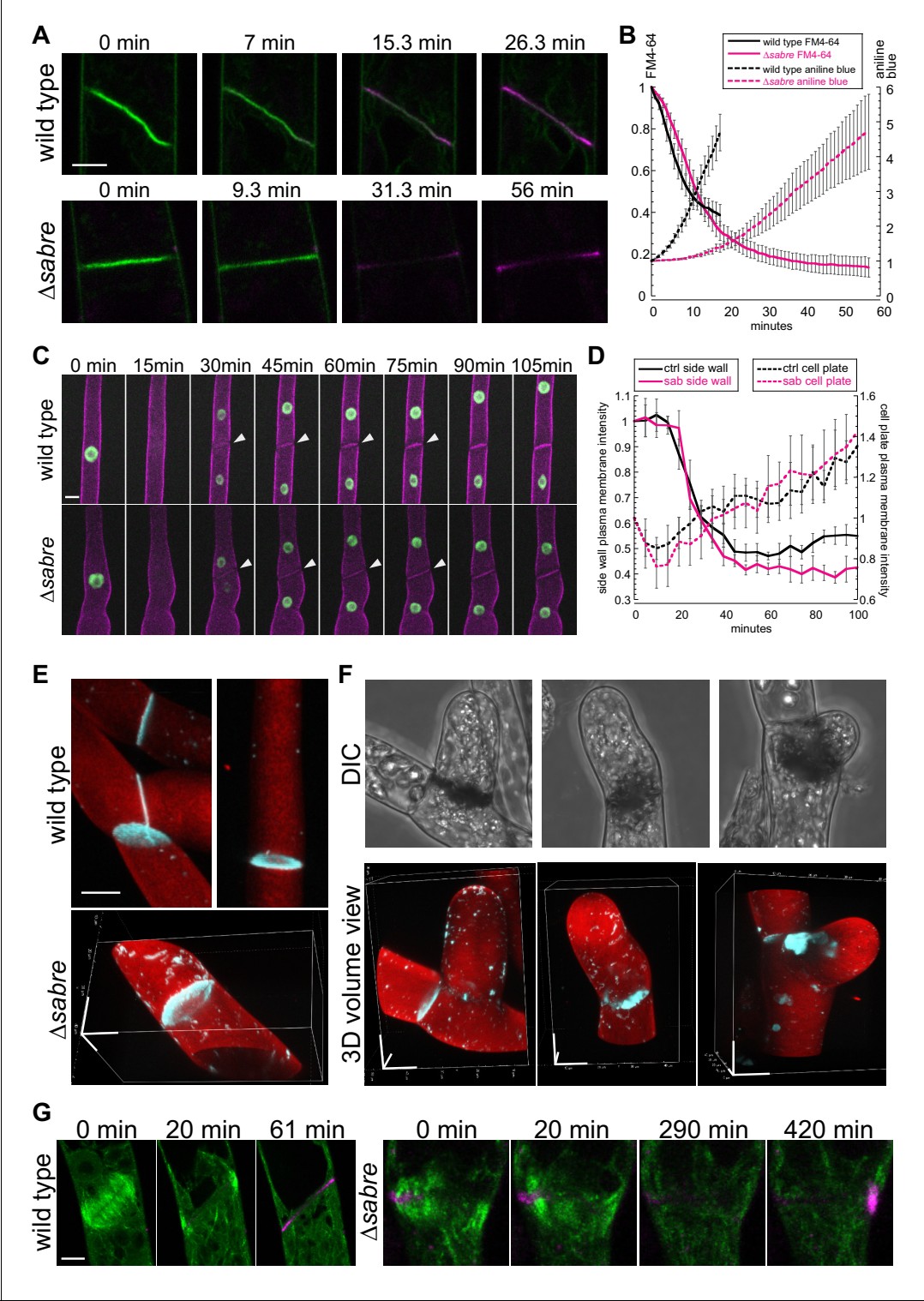

**Figure 8.** Aniline blue deposition is altered in Δ*sabre*. (**A**) Time lapse of FM4-64 (green) and aniline blue (magenta) showing aniline blue staining the cell plate after FM4-64 staining diminishes. Also see *Video 13*. (**B**) Quantification of FM4-64 and aniline blue at the developing cell plate over time. Intensity is measured by drawing a 15-pixel wide curved line along the developing cell plate and measuring the mean intensity value. Each data point is the average of N = 6 cells for each category. Intensity is normalized to the starting point of the time lapse – the moment when FM4-64 is the strongest and no aniline blue stain accumulates. Error bars, standard error of the mean. (**C**) Time lapse of membrane marker SNAP-TM-mCherry (magenta) and nuclear marker NLS-GFP-GUS (green) during cell division. A 0 min frame depicts the last time point (within 5 min) before nuclear envelope breaks down. Also see *Video 11*. (**D**) Quantification of signal intensity in (**C**). A region of interest of 6–10 μm² was manually drawn at the side wall where

*Figure 8 continued on next page*

*Figure 8 continued*

membrane fusion occurs and at the middle of the cell plate, respectively, and mean intensity was measured for those two regions of interests and plotted against time. Each data point is the average of N = 6 cells for each category. Error bars, standard error of the mean. (E) Example 3D images of normal mature cell plates stained with aniline blue (cyan) and fast scarlet (F2B) (red) to outline the cell wall. (F) Examples of different aniline blue staining patterns in Δ*sabre* mutants containing brown material. Left, weak stain; middle, partial stain; and right, large chunk near the cell plate. Scale bars, 10 μm. (G) Time lapse showing aniline blue staining (magenta) in a wild type cell and a Δ*sabre* cell with brown material. Images are maximum projections of deconvolved confocal Z-stacks. Microtubules labeled with GFP-tubulin are shown in green. Scale bar, 5 μm. Also see *Video 14*.

The online version of this article includes the following source data for figure 8:

**Source data 1.** Measurement of FM4-64 and aniline blue intensities.

suggesting that SABRE's influence on the ER might alter ER secretory function. In contrast to defects in COPII function, which generally reduces secretion, Δ*sabre* defects appear to specifically influence a subset of secretory cargo. We discovered that delivery of the plasma membrane protein SNAP-TM-mCherry, which was affected in COPII mutants (*Chang et al., 2020*), was unaffected in Δ*sabre*. In a surprising connection, a study in *Drosophila* discovered hobbit, a protein that the authors report is conserved broadly across eukaryotes (*Neuman and Bashirullah, 2018*). *SABRE* is the putative plant hobbit homolog albeit with significant sequence divergence. Even with the vast evolutionary distance between flies and plants, hobbit localizes to the ER when overexpressed in *Drosophila* and hobbit mutants are stunted similar to *sabre* null mutants in both Arabidopsis and *P. patens*. In *Drosophila*, mutants in hobbit accumulated proteins required for membrane fusion in endosomal compartments and were defective specifically in insulin secretion, manifesting in stunted growth. If hobbit and SABRE function are conserved, then in plants SABRE may regulate a subset of secretory cargos critical for cell expansion and division.

Alternatively, SABRE might influence the composition of regions of the ER membrane. Altered distribution or activity of ER resident membrane proteins, such as ethylene receptors (*Ji and Guo, 2013*; *Yang et al., 2015*), could impact growth and development. Previous studies in Arabidopsis provide a link between ethylene, a gaseous phytohormone involved in a variety of developmental processes and stress responses (*Binder, 2020*; *Binder and Eric Schaller, 2017*), and SABRE since inhibition of ethylene biogenesis partially rescued the *sabre* mutant in Arabidopsis (*Aeschbacher et al., 1995*; *Yu et al., 2012*). To distinguish between altered ethylene responses versus secretory defects, comparative RNA-seq and proteomic studies in Δ*sabre* versus wild type could provide future research directions to narrow down SABRE's influence on ER function. Another intriguing possibility is based on SABRE's impact on callose deposition. Perhaps during cytokinesis SABRE is recruited to the cell plate membrane via ER–plasma membrane contact sites, and there SABRE regulates callose synthase activity ensuring uniform deposition of callose.

Our results have revealed that the ER does not simply repopulate at the daughter plasma membranes during cell division. Instead, the ER, together with SABRE, is critical for cell plate maturation and is involved in regulating callose deposition. Furthermore, the ER via SABRE ultimately impacts cell expansion and nuclear migration.

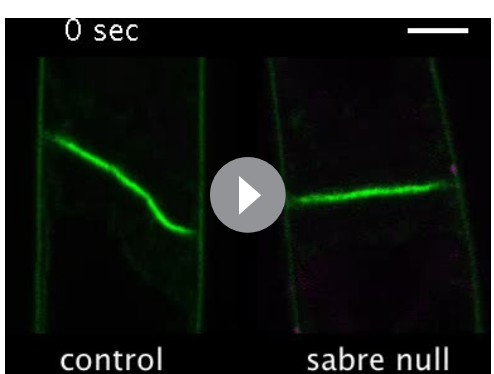

**Video 13.** Aniline blue and FM4-64 staining at developing cell plate. Green, FM4-64; magenta, aniline blue. Images are medial sections from dividing cells. Scale bar, 5 μm. Video is playing at 8 fps.
https://elifesciences.org/articles/65166#video13

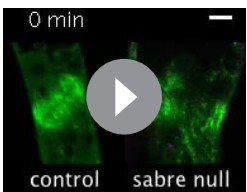

**Video 14.** Aniline blue staining during normal division and cytokinesis failure in Δ*sabre*. Images are maximum projections of deconvolved confocal Z-stacks taken every 10 min. Green, microtubule; magenta, aniline blue. Scale bar, 10 μm. Video is playing at 5 fps.
https://elifesciences.org/articles/65166#video14

Future studies will investigate the interactions between SABRE and ER-localized proteins involved in protein trafficking, ethylene sensing, and cell wall synthesis. Results from these studies will importantly unravel whether the ER influence on cell division, cell expansion, and nuclear migration results from defective secretion or altered ER membrane composition and function.

# Materials and methods

## Key resources table

| Reagent type (species) or resource | Designation | Source or reference | Identifiers | Additional information |
|---|---|---|---|---|
| Gene (*Physcomitrium patens*) | *SABRE* | Phytozome | Pp3c12_12980 | Gene of interest in this study |
| Other | Calcofluor white | Sigma Aldrich | 18909 | 0.1 mg/mL dissolved in Hoagland's media |
| Other | Propidium iodide | Sigma Aldrich | 81845 | 15 µg/mL dissolved in Hoagland's media |
| Other | FM-4-64 | Invitrogen | T3166 | 15 µM dissolved in Hoagland's media |
| Other | Aniline blue | Fisher Scientific | 28631-66-5 | 20 µg/mL dissolved in Hoagland's media |
| Other | Fast Scarlet | Sigma Aldrich | R320919 | 50 µg/mL dissolved in Hoagland's media |
| Sequenced-based reagent | DC65 | This paper | PCR primers | caccATGGAGGTTACACCTGAC |
| Sequenced-based reagent | DC164 | This paper | Protospacer primer | ccatTCAGTGCGCGAGTAAGCTTC |
| Sequenced-based reagent | DC164 | This paper | Protospacer primers | aaacGAAGCTTACTCGCGCACTGA |
| Sequenced-based reagent | DC175 | This paper | PCR primers | GGGGACAAGTTTGTACAAAAAAGCA GGCTTAGTGATTGAGCAACAGCTATTGC |
| Sequenced-based reagent | DC176 | This paper | PCR primers | GGGGACAACTTTGTATAGAAAAGTT GGGTGGAACCCTGCTGGCTATC |
| Sequenced-based reagent | DC177 | This paper | PCR primers | GGGGACAACTTTGTATAATAAAGTT GTAGCTTCCGGTTAGCTGGT |
| Sequenced-based reagent | DC168 | This paper | PCR primers | GGGGACCACTTTGTACAAGAAAGC TGGGTTCTGCTGGATACAGTGAGATG |
| Sequenced-based reagent | DC265 | This paper | Sequencing primes | TGTAATTATTCCAGAAGTGTTAGG |
| Sequenced-based reagent | DC191 | This paper | Sequencing primers | CAAGATAACCTCCACATCCG |
| Sequenced-based reagent | DC266 | This paper | PCR primers | GCAGAAAGAATTGAGGTTGG |
| Sequenced-based reagent | DC267 | This paper | PCR primers | CCGATCAGAATGATCAACAAG |
| Sequenced-based reagent | DC314 | This paper | Protospacer primers | ccatGGCCGTGACTCTCCCCTCTG |
| Sequenced-based reagent | DC315 | This paper | Protospacer primers | aaacCAGAGGGGAGAGTCACGGCC |
| Sequenced-based reagent | DC316 | This paper | Protospacer primers | ccatCGATACCCCATCAGCTTACG |
| Sequenced-based reagent | DC317 | This paper | Protospacer primers | aaacCGTAAGCTGATGGGGTATCG |
| Sequenced-based reagent | DC322 | This paper | PCR primers | GGGGACAAGTTTGTACAAAAAAGCAGG CTTACTAGGACGCTGGGCTAAG |

*Continued on next page*

*Continued*

| Reagent type (species) or resource | Designation | Source or reference | Identifiers | Additional information |
|---|---|---|---|---|
| Sequenced-based reagent | DC323 | This paper | PCR primers | GGGGACAACTTTGTATAGAAAAGTTGG GTGGAATCCAACACTTCAGAGGC |
| Sequenced-based reagent | DC324 | This paper | PCR primers | GGGGACAACTTTGTATAATAAAGTTGTAA TGGAGGTTACACCTGACAAAT |
| Sequenced-based reagent | DC325 | This paper | PCR primers | GGGGACCACTTTGTACAAGAAAGCTGGGTT GAACCCTGCTGGCTATC |
| Sequenced-based reagent | DC326 | This paper | PCR primers | GGGGACAAGTTTGTACAAAAAAGCAGGC TTAAGTTCAAGGATAAGTTACCCGC |
| Sequenced-based reagent | DC327 | This paper | PCR primers | GGGGACAACTTTGTATAGAAAAGTTGGG TGATCCAAGTTCTCGTAAGCTGATG |
| Sequenced-based reagent | DC328 | This paper | PCR primers | GGGGACAACTTTGTATAATAAAGTT GTACAGCAATAACCATCCAGTTTTGTA |
| Sequenced-based reagent | DC329 | This paper | PCR primers | GGGGACCACTTTGTACAAGAAAGC TGGGTTGCTGTGAAACAGTGAGGTC |
| Sequenced-based reagent | DC403 | This paper | PCR primers | GGTCACGTGCTTGCAT |
| Sequenced-based reagent | DC404 | This paper | PCR primers | CGTCTTTGAGTCGTTGAAAAC |
| Sequenced-based reagent | DC405 | This paper | PCR primers | ACATACATTCTGTAGCACTCAC |
| Sequenced-based reagent | DC406 | This paper | PCR primers | GAACAAGTGATTTGGTTCCTG |
| Sequenced-based reagent | DC1059 | This paper | PCR primers | TTCTTGTTTCACGACAGGG |
| Sequenced-based reagent | DC473 | This paper | Sequencing primers | CCAAGAGGTCAGCCTTTC |
| Sequenced-based reagent | DC474 | This paper | Sequencing primers | GACGTGAAGGACCAAAGC |
| Sequenced-based reagent | DC475 | This paper | Sequencing primers | GCATACGAAACAATACCGATG |
| Sequenced-based reagent | DC625 | This paper | PCR primers | GGGGACAACTTTTCTATACAAAGTT GTAGGATCCATGGTGAGTAAAGGCGAGG |
| Sequenced-based reagent | DC626 | This paper | PCR primers | GGGGACAACTTTATTATACAAAGTT GTTTACTTATACAATTCGTCCATACCCATC |
| Sequenced-based reagent | DC627 | This paper | PCR primers | GGGGACAACTTTATTATACAAAGTT GTCTTATACAATTCGTCCATACCCATC |
| Sequenced-based reagent | DC632 | This paper | Sequencing primers | CAATGGTTGACGGATCA |
| Sequenced-based reagent | DC633 | This paper | Sequencing primers | TTAGAACGGCACCAATCA |
| Sequenced-based reagent | DC634 | This paper | Sequencing primers | GGCTATGGTAGATGGCAGT |
| Sequenced-based reagent | DC635 | This paper | Sequencing primers | CTACGGCACCAATCGGCA |
| Sequenced-based reagent | DC791 | This paper | PCR primers | TAGCGTGGATCCATGGT AAGCAAAGGAGAGGAGG |
| Sequenced-based reagent | DC792 | This paper | PCR primers | TAGCGTAGATCTCTTGT ATAACTCATCCATGCCC |
| Sequenced-based reagent | DC793 | This paper | PCR primers | TAGCGTGGATCCATGG TGAGTAAAGGCGAGG |
| Sequenced-based reagent | DC794 | This paper | PCR primers | TAGCGTAGATCTCTTATAC AATTCGTCCATACCCATC |

*Continued on next page*

*Continued*

| Reagent type (species) or resource | Designation | Source or reference | Identifiers | Additional information |
|---|---|---|---|---|
| Sequenced-based reagent | DC818 | This paper | PCR primers | GGGGACAACTTTTCTATACAAAGTT GGGCTAGAGATAATGAG CATTGCATGTCTAAG |
| Sequenced-based reagent | DC819 | This paper | PCR primers | GGGGACAACTTTATTATACAAAGTT GTGCAGAAGTAACACCAAACAACAGG |
| Sequenced-based reagent | DC1055 | This paper | Protospacer primers | ccatGTTGCCAAGTTCGCCGGGCT |
| Sequenced-based reagent | DC1056 | This paper | Protospacer primers | aaacAGCCCGGCGAACTTGGCAAC |
| Sequenced-based reagent | DC1057 | This paper | Protospacer primers | ccatGTCATGGAAGGTTCGGTCAA |
| Sequenced-based reagent | DC1058 | This paper | Protospacer primers | aaacTTGACCGAACCTTCCATGAC |
| Recombinant DNA reagent | pMH-SAB-stop plasmid | This paper | BP-1301 | Materials and methods, distributed by Bezanilla Lab |
| Recombinant DNA reagent | pGEM-SAB-stop plasmid | This paper | BP-1302 | Materials and methods, distributed by Bezanilla Lab |
| Recombinant DNA reagent | pMH-SAB-C plasmid | This paper | BP-1303 | Materials and methods, distributed by Bezanilla Lab |
| Recombinant DNA reagent | pENTR R4R3 Cterm BamHI 1xSc_mNeon plasmid | This paper | BP-1304 | Materials and methods, distributed by Bezanilla Lab |
| Recombinant DNA reagent | pENTR R4R3 Cterm BamHI 2xPp_Sc_mNeon plasmid | This paper | BP-1305 | Materials and methods, distributed by Bezanilla Lab |
| Recombinant DNA reagent | pENTR R4R3 Cterm BamHI 3xSc_Pp_Sc_mNeon plasmid | This paper | BP-1306 | Materials and methods, distributed by Bezanilla Lab |
| Recombinant DNA reagent | pENTR-R4R3-Ubiquitin-pro plasmid | This paper | BP-1307 | Materials and methods, distributed by Bezanilla Lab |
| Recombinant DNA reagent | pGEM-SAB-3GFP plasmid | This paper | BP-1308 | Materials and methods, distributed by Bezanilla Lab |
| Recombinant DNA reagent | pGEM-SAB-3mNG plasmid | This paper | BP-1309 | Materials and methods, distributed by Bezanilla Lab |
| Recombinant DNA reagent | pMH-SAB-N plasmid | This paper | BP-1310 | Materials and methods, distributed by Bezanilla Lab |
| Recombinant DNA reagent | pGEM-Ubipro-SAB plasmid | This paper | BP-1311 | Materials and methods, distributed by Bezanilla Lab |
| Recombinant DNA reagent | pMH-mRuby2-2ps plasmid | This paper | BP-1312 | Materials and methods, distributed by Bezanilla Lab |
| Recombinant DNA reagent | pTZ-SP-mCherry-KDEL plasmid | This paper | BP-1313 | Materials and methods, distributed by Bezanilla Lab |
| Strain, strain background (*Physcomitrium patens*) | Wild type | Gransden 2011 | BL-1 | |
| Strain, strain background (*Physcomitrium patens*) | GFP-tubulin | *Wu and Bezanilla, 2018* | BL-164 | |
| Strain, strain background (*Physcomitrium patens*) | Lifeact-GFP | *van Gisbergen et al., 2012* | BL-546 | |
| Strain, strain background (*Physcomitrium patens*) | SP-GFP-KDEL | *Chang et al., 2020* | BL-541 | |

*Continued on next page*

*Continued*

| Reagent type (species) or resource | Designation | Source or reference | Identifiers | Additional information |
|---|---|---|---|---|
| Strain, strain background (*Physcomitrium patens*) | NLS-GFP-GUS/ SNAP-TM-mCherry | *van Gisbergen et al., 2018* | BL-136 | |
| Strain, strain background (*Physcomitrium patens*) | mCherry-tubulin | *Burkart et al., 2015* | BL-159 | |
| Strain, strain background (*Physcomitrium patens*) | Lifeact-mRuby | *van Gisbergen et al., 2020* | BL-328 | |
| Strain, strain background (*Physcomitrium patens*) | Δ*sab*/wild type | This paper | BL-650 | *Supplementary file 1*, distributed by Bezanilla Lab |
| Strain, strain background (*Physcomitrium patens*) | Δ*sab*/GFP-tubulin | This paper | BL-653 | *Supplementary file 1*, distributed by Bezanilla Lab |
| Strain, strain background (*Physcomitrium patens*) | Δsab/Lifeact-GFP | This paper | BL-654 | *Supplementary file 1*, distributed by Bezanilla Lab |
| Strain, strain background (*Physcomitrium patens*) | Δsab/ER-GFP | This paper | BL-656 | *Supplementary file 1*, distributed by Bezanilla Lab |
| Strain, strain background (*Physcomitrium patens*) | Δsab/NLS-GFP/ SNAP-mCh | This paper | BL-658 | *Supplementary file 1*, distributed by Bezanilla Lab |
| Strain, strain background (*Physcomitrium patens*) | SAB-3GFP | This paper | BL-660 | *Supplementary file 1*, distributed by Bezanilla Lab |
| Strain, strain background (*Physcomitrium patens*) | SAB-3mNG/mCh-tub | This paper | BL-661 | *Supplementary file 1*, distributed by Bezanilla Lab |
| Strain, strain background (*Physcomitrium patens*) | SAB-3mNG/LAmR | This paper | BL-662 | *Supplementary file 1*, distributed by Bezanilla Lab |
| Strain, strain background (*Physcomitrium patens*) | OE-SAB-3mNG/mCh-tub | This paper | BL-664 | *Supplementary file 1*, distributed by Bezanilla Lab |
| Strain, strain background (*Physcomitrium patens*) | OE-SAB-3mNG/LAmR | This paper | BL-667 | *Supplementary file 1*, distributed by Bezanilla Lab |
| Strain, strain background (*Physcomitrium patens*) | SAB-3mNG/ER-mCherry | This paper | BL-668 | *Supplementary file 1*, distributed by Bezanilla Lab |
| Strain, strain background (*Physcomitrium patens*) | OE-SAB-3mNG/ ER-mCherry | This paper | BL-669 | *Supplementary file 1*, distributed by Bezanilla Lab |
| Strain, strain background (*Physcomitrium patens*) | Δsab/ER-mCherry | This paper | BL-670 | *Supplementary file 1*, distributed by Bezanilla Lab |

## Plasmid construction

All genomic modifications in this study were performed using CRISPR-Cas9-mediated HDR. In brief, two plasmids were generated: a CRISPR plasmid that contains the protospacer(s) and Cas9 ultimately generating the double-stranded break(s) at the designated genomic site(s) and a homology plasmid that provides the template in addition to the sequence being inserted (knockout cassette, fluorescent protein sequence, or promoter) for DNA repair. The two plasmids were co-transformed into moss protoplasts and transformants were regenerated from single protoplasts. All plasmids were constructed using the methods and modular vectors described in *Mallett et al., 2019*. Primers

used to generate these plasmids along with the corresponding plasmid products and primers used for subsequent genotyping are listed in *Supplementary file 1*. Plasmids were transformed into a variety of moss lines stably expressing fluorescently labeled markers. These lines and the corresponding new lines generated in this study are listed in *Supplementary file 1*. Note that for generating SABRE-3mNG tag we initially transformed pMH-SAB-C and pGEM-SAB-3mNG into moss lines expressing Lifeact-mRuby and mCherry-tubulin. To introduce ER labeling, we used CRISPR-mediated HDR to swap Lifeact-mRuby into mCherry-KDEL (*Supplementary file 1*). All other fluorescent labeled lines are as described before: GFP-tubulin (*Wu and Bezanilla, 2018*), Lifeact-GFP (*van Gisbergen et al., 2012*), SP-GFP-KDEL (*Chang et al., 2020*), NLS-GFP-GUS/SNAP-TM-mCherry (*van Gisbergen et al., 2018*, p. 10), mCherry-tubulin (*Burkart et al., 2015*), and Lifeact-mRuby (*van Gisbergen et al., 2020*).

To generate the three tandem mNeonGreen (3mNG) tag, we amplified mNG codon optimized for budding yeast with DC625 and DC626 (*Supplementary file 1*) incorporating a BamHI site just upstream of the ATG. This PCR product was introduced into pDONR221-P4rP3r using BP reaction (Invitrogen) generating pENTR-R4R3-mNG. A second mNG codon optimized for *P. patens* was amplified without the stop codon and incorporating BamHI upstream and BglII downstream. This product was ligated into pENTR-mNG in-frame upstream using the BamHI site to create pENTR-R4R3-2XmNG. The third mNG codon optimized for budding yeast was similarly amplified and ligated into pENTR-R4R3-2XmNG. The resulting pENTR R4R3 Cterm BamHI 3xSc_Pp_Sc_mNeon plasmid (*Supplementary file 1*) was used to create the homology plasmid pGEM-SAB-3mNG according to *Mallett et al., 2019*. Similarly, to knock-in the stronger constitutive maize ubiquitin promoter, pENTR-R4R3-Ubiquitin-pro was generated by inserting the ubiquitin promoter into pDONR221P4rP3r with BP reaction (*Supplementary file 1*), and the resulting entry clone was used to make the homology plasmid for moss transformation.

## Plant culture and transformation

Moss tissue was cultured on $PpNH_4$ medium (1.03 mM $MgSO_4$, 1.86 mM $KH_2PO_4$, 3.3 mM Ca$(NO_3)_2$, 2.72 mM $(NH_4)_2$-tartrate, 45 µM $FeSO_4$, 9.93 µM $H_3BO_3$, 220 nM $CuSO_4$, 1.966 µM $MnCl_2$, 231 nM $CoCl_2$, 191 nM $ZnSO_4$, 169 nM KI, and 103 nM $Na_2MoO_4$) supplied with 0.7% agar, plated on Petri dishes. After propagation by blending, 5–7-day-old tissue was protoplasted and then transformed as previously described (*Liu and Vidali, 2011*). For all the HDR transformations, 7.5 µg of the CRISPR/protospacer plasmid and 7.5 µg of the homology plasmid were co-transformed into 150 µL protoplasts at a concentration of 2,000,000 protoplasts/mL. Transformed protoplasts were resuspended in liquid plating medium ($PpNH_4$ plus 8.5% mannitol and 10 mM $CaCl_2$), plated and regenerated on PRM-B media ($PpNH_4$ plus 6% mannitol and 10 mM $CaCl_2$) with 0.8% agar. A layer of cellophane was placed on top of the PRM-B plates, and protoplasts were plated on top of the cellophane. After 4 days on PRM-B, the cellophane was transferred to $PpNH_4$ supplied with antibiotic for selection. Then, 15 µg/mL hygromycin was used for selection of transformed protoplasts. Plants were grown on selection for a week before moving to $PpNH_4$ media for subsequent culturing and genotyping.

Growth assays were used to quantify protonemal area. Tissue regenerated from protoplasts was used to synchronize plant growth. Protoplasts were isolated, plated, and regenerated as described above. After 4 days on PRM-B, they were transferred to $PpNH_4$ and allowed to grow for another 3 days. Seven days after protoplasting, plants were imaged with a Nikon SMZ25 stereomicroscope equipped with a color camera (Nikon digital sight DS-Fi2). Plants were transferred from the plate to a slide and stained with 0.1 mg/mL calcofluor. Calcofluor fluorescence was imaged with a violet filter cube (excitation 420/25, dichroic 455, emission 460 longpass). Subapical cell length was measured manually using these images. Quantification of plant area was carried out using the methods modified from *Vidali et al., 2007*. In brief, colored images were converted to a single red color image. Single plants were selected and highlighted by cropping and thresholding above a certain intensity value. Plant area was calculated based on the thresholded images. For each experiment, plant area was normalized to the average area of control plants.

## Cloning transcript sequences

Total mRNA was extracted from fresh tissue of both wild type and Δsabre. Total cDNA was generated with oligo-dT primers using extracted mRNA as the template. Primers DC65 and DC191 (*Supplementary file 1*) were used to amplify the CDS encoding the N terminal portion of the SABRE protein (*Figure 1—figure supplement 1A*). Amplified fragments were cloned into pGEM-T Easy (Promega) after A tailing. The cloned fragments were then confirmed with Sanger sequencing.

## Brightfield microscopy

Brightfield time-lapse microscopy was performed using a Nikon Ti microscope equipped with a 0.8 NA ×20 objective. Plants were cultured in continuous light in PDMS microfluidic devices with liquid Hoagland's medium as previously described (*Bascom et al., 2016*). For imaging protonemal tissue, ground tissue was loaded into microfluidic devices and allowed to grow at least 2–3 days before imaging. Gametophores emerged from protonemata naturally 2–3 weeks after loading ground tissue. Multiple XY positions were acquired, and a either a single focal plane or a Z-stack was acquired for each position. Between each time point, white light remained on to provide light for plant growth. Mono-color brightfield images were acquired with Nikon DS-Qi2 camera. Colored brightfield images were acquired with Nikon DS-Vi1 camera. Extended-depth-of-focus (EDF) images were created for the Z-stacks with NIS-Elements (Nikon). Protonemata growth rate was measured by manually tracing the growing tip during active growing period in NIS-Elements software.

## Laser scanning confocal fluorescence microscopy

For short-term imaging, moss protonemal tissue was mounted on an agar pad on a slide, submerged in Hoagland's medium, and sealed with a coverslip. For protonemata staining, FM4-64 (15 μM), aniline blue (20 μg/mL), and Fast Scarlet (50 μg/mL) were dissolved in Hoagland's medium used for tissue mounting. For imaging tissue regenerated from protoplasts, regenerating protoplasts were removed from the cellophane, loaded into microfluidic devices, and immediately imaged. To synchronize cell divisions, ground tissue was loaded into microfluidic devices and allowed at least 4 days to grow. The microfluidic device was placed in far red light for 3–4 days before exposure to white light and imaging. Confocal imaging was performed using a Nikon A1R laser scanning confocal with a 1.3 NA ×40 or 1.49 NA ×60 oil immersion objective (Nikon). Laser illumination at 405 nm was used for exciting aniline blue dye, 488 nm was used for exciting mNeonGreen, GFP, and chlorophyll autofluorescence; 561 nm for mRuby2, mCherry, FM4-64, Fast Scarlet, and propidium iodide. Emission filters were 525/50 nm for mNeonGreen/GFP and aniline blue; 595/50 nm for mRuby2, mCherry, FM4-64, Fast Scarlet, and propidium iodide. For chlorophyll autofluorescence emission, light passed through a long-pass filter allowing wavelengths larger than 640 nm to pass. Image acquisition was controlled by Nikon NIS-Elements software (Nikon). In between each time point, transmitted white light was on providing light for plant growth. 3D reconstruction was done using 3D volume viewer with maximum projection rendering method in NIS-Elements, and contrast for slices at different Z positions was adjusted individually to compensate for loss of signal in tissue further away from the objective using 3D lookup table function. Deconvolution was carried out with NIS-Elements (Nikon) with the 2D deconvolution default settings.

To quantify cell size in gametophores, mature phyllids were removed from gametophore and mounted in a droplet of staining solution (15 μg/mL propidium iodide dissolved in liquid Hoagland's medium) between a slide and coverslip. Confocal images were captured for quantification. In Fiji (*Schindelin et al., 2012*), fluorescent images were processed using enhance contrast, subtract background, and smooth and median filter. The processed images were then converted to a binary mask and put through binary process, Close>Dilate>Close>Skeletonize>Dilate, to outline the edges of the cells. Images were then inverted to highlight the cell area and subsequently quantified using the analyze particle function. After quantification, incorrect cells (fused or broken) were manually removed.

## Variable angle epifluorescence microscopy

VAEM microscopy was performed using a Nikon Ti-E inverted microscope equipped with a TI-TIRF-PAU illuminator, using a Nikon 1.49 NA ×100 oil immersion TIRF objective. Also, 6–8-day-old plants regenerated from protoplasts were mounted between a coverslip and an agar pad on top of a slide,

prepared right before imaging. GFP and mNeonGreen were illuminated with a 488 nm laser, while mRuby2 and mCherry were excited with a 561 nm laser, the emission passed through a 525/50 filter for GFP/mNeonGreen and 610/75 for mRuby2/mCherry. Images were simultaneously captured with two Andor 897 EMCCD cameras. Image acquisition was controlled by Nikon NIS-Elements software. All data was processed with enhanced contrast (0.1% pixel saturation), subtract background and smoothing in Fiji using default settings.

To analyze SABRE co-localization with either ER, actin, or microtubules, Pearson's correlation coefficients were calculated using established algorithms in the NIS-Elements software package. To measure co-localization using an intensity-independent method, we measured the fraction SABRE area overlapping with the either ER, actin, or microtubules. Time-lapse images were processed as described above and as described in *Figure 4—figure supplement 3C*. Specifically, using Fiji, the SABRE channel was thresholded with the MaxEntropy method uniformly to select area with intensity between 36,000 and 65,535, then converted to a binary mask. The total SABRE area was acquired by measuring the white area of this mask. The ER, actin, or microtubule channel was filtered using unsharp mask with a radius = 3 and mask = 0.6, thresholded with the Otsu method between 28,000 and 65,535, and converted to a binary mask. The binary mask image was skeletonized and dilated to create the ER, actin, or microtubule skeleton. The skeletonized image was then flipped vertically. Then original or flipped skeleton was subtracted from the SABRE binary mask to extract the proportion of SABRE area not overlapping with ER, actin, or microtubules. These processes were done for the first 50 frames of the time-lapse video with 120 ms interval, and the average of each frame was calculated as a data point for each cell. The area of SABRE that did overlap was derived from the above number, which equals to one minus the fraction that did not overlap.

Cortical actin and microtubule dynamics were quantified by measuring the decay of the correlation coefficient over time, as previously described (*Vidali et al., 2010*). Briefly, two frames within the time-lapse video that were 1, 2, 3 … to N (largest frame number present in the video) frames apart were paired, and the correlation coefficient between every pixel position in those two images was calculated. The numbers for the same frame numbers apart were then averaged between different cells to generate the data point, and standard error of the means was calculated to generate the error bar. The quicker the decrease in the correlation coefficient as the temporal spacing between the frames increased indicates more dynamic movement of the fluorescence signal.

## Fluorescence recovery after photobleaching

Phragmoplast microtubule photobleaching experiments were conducted using a Nikon A1R laser scanning confocal microscope with 1.49 NA ×60 oil immersion objective. Actively growing plant tissue was mounted on an agar pad between the slide and coverslip, then imaged immediately with a 488 nm laser to identify actively dividing cells at the phragmoplast expansion stage. A 3 µm × 3 µm square region of interest (ROI) was placed in the center of phragmoplast, ensuring the entire ROI was filled with phragmoplast microtubules. Photobleaching was carried out using a 405 nm laser at 10% power for 1 s, after six frames (5 s) of normal imaging. Imaging continued after photobleaching for 2 min to capture fluorescence recovery. The average intensity in the ROI was measured for each frame, then normalized to the average of the value from the first six frames.

## Nuclear migration trajectory analysis

A moss line with the nuclear marker NLS-GFP-GUS and the plasma membrane marker SNAP-TM-mCherry (*van Gisbergen et al., 2018*) was imaged with time-lapse confocal microscopy. A Z-stack was taken every 5 min for the apical cells of several filaments. A segmented line was drawn manually along the axis of growth to generate the kymograph. To make the trends on the kymograph easier to label (*Figure 6C*), we stretched the kymograph image by increasing the Y axis twofold. To measure the basal nuclear position, using the straight-line tool in Fiji we measured the distance between the middle of the cell plate and the basal edge of the nucleus when it was closest to the cell plate in the kymograph. The distance was divided by the cell length at the same time point, to generate the relative basal nuclear position. To analyze nuclear migration, we isolated movie fragments of nuclear apical migrating periods, tracked the nuclear GFP signal with the TrackMate plugin in Fiji (*Tinevez et al., 2017*), with a spot diameter of 10 µm. Displacement, distance, and velocity were calculated. Nuclear migration displacement was defined as the straight-line distance between the initial

and final positions of the nucleus. Total migration distance was calculated as the sum of displacement between each frame. Average instantaneous velocity was calculated between each frame.

## Acknowledgements

We thank Charles Barlowe lab (Dartmouth College) and Gohta Goshima lab (Nagoya University) for sharing mNeonGreen gene template with two different codon usages. We thank Ann Lavanway from Dartmouth College for helping with VAEM (TIRF) microscopy. The acquisition of the TIRF microscope at Dartmouth College was funded by an NIH S10, grant number 1S10OD018046-01. This work was supported by Dartmouth College, a grant from the National Science Foundation (MCB-1715785 to MB) and the John H Copenhaver Jr and William H Thomas MD 1952 Award from Dartmouth Molecular and Cellular Biology program.

## Additional information

### Funding

| Funder | Grant reference number | Author |
| --- | --- | --- |
| National Science Foundation | MCB-1715785 | Magdalena Bezanilla |
| Dartmouth College | John H. Copenhaver Jr. and William H. Thomas MD 1952 Award | Xiaohang Cheng |

The funders had no role in study design, data collection and interpretation, or the decision to submit the work for publication.

### Author contributions

Xiaohang Cheng, Magdalena Bezanilla, Conceptualization, Formal analysis, Supervision, Funding acquisition, Validation, Investigation, Visualization, Methodology, Writing - original draft, Project administration, Writing - review and editing

### Author ORCIDs

Magdalena Bezanilla (iD) https://orcid.org/0000-0001-6124-9916

### Decision letter and Author response

Decision letter https://doi.org/10.7554/eLife.65166.sa1
Author response https://doi.org/10.7554/eLife.65166.sa2

## Additional files

### Supplementary files

• Supplementary file 1. Supplemental Table 1. Primers used in this study and the plasmid constructs they are used to generate, respectively. Supplemental Table 2. Plasmids used to transform moss and the lines generated from those transformations. Supplemental Table 3. One-way ANOVA for *Figure 1C*. Supplemental Table 4. One-way ANOVA for *Figure 2E*. Supplemental Table 5. One-way ANOVA for *Figure 2F*. Supplemental Table 6. One-way ANOVA for *Figure 4E*. Supplemental Table 7. One-way ANOVA for *Figure 4—figure supplement 2A*. Supplemental Table 8. One-way ANOVA for *Figure 4—figure supplement 3D*, left graph. Supplemental Table 9. One-way ANOVA for *Figure 4—figure supplement 3D*, right graph.

• Transparent reporting form

### Data availability

All data generated or analyzed during this study are included in the manuscript and supporting files.

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
