## [Decision Letter]

**Acceptance summary:**

The *SABRE* protein family is critical for growth in plants and animals, but the cellular mechanisms by which these proteins act were previously unknown. Functional genetic and live cell imaging approaches in the moss Physcomitrium enabled the authors to gather data that change our current view of *SABRE* activity. Rather than a role regulating the cytoskeleton (specifically microtubules), they show that *SABRE* influences membrane trafficking. Additionally this work implicates the ER in ensuring cell plate integrity in plant cell divisions.

**Decision letter after peer review:**

Thank you for submitting your article "*SABRE* populates ER domains essential for cell plate maturation and cell expansion influencing cell and tissue patterning" for consideration by *eLife*. Your article has been reviewed by three peer reviewers, one of whom is a member of our Board of Reviewing Editors, and the evaluation has been overseen by Christian Hardtke as the Senior Editor. The reviewers have opted to remain anonymous.

Essential revisions:

1) The work here describes new localization and roles for a plant *SABRE* protein, and attempts to define the cellular mechanisms that underlie the defects observed upon loss of the protein in Physcomitrium. While it was appreciated that the authors touched on many different aspects of potential *SABRE* functions, the result is a somewhat diffuse manuscript, and the expectation that you would make sense of the phenotypes by explaining molecular functions of the *SABRE* protein. There are several ways to address this. One solution would be to investigate one hypothesis for *SABRE* function more in depth. For example, reviewer 1 suggests testing whether the author's hypothesis that *SABRE* may be the link to motors for nuclear migration. The other acceptable route would be to modify the text to highlight the findings in a way that people would appreciate the careful and comprehensive cellular analysis rather than expecting the molecular mechanism of *SABRE*.

2) The colocalization of *SABRE* with the ER are based on Pearson correlation coefficients (relative to other compartments) but details of the calculation are missing from the Materials and methods. Because localization is a major finding of the paper, and is in contrast to previous work, additional details on how this was measured should be included. Specific places to revise include: (1) “the correlation coefficient between images over all possible temporal spacings” to show actin/microtubule dynamics. Please explain more about the quantification data in Figure 2—figure supplement 1C and D in the main text. (2) The authors conclude that *SABRE* is ER-localized, but since *SABRE-3mNG* is detected in punctae, whereas ER-mCherry is shown as tubules and sheets, morphologically they are different. More evidence (biochemical or cell biological evidence) is required here to support the conclusion of ER-localization. In general, the authors should make sure their colocalization claims here are supported, as historically, there may have been over-confidence in imaging-based analyses.(see Aaron JS, Taylor AB, Chew TL. Image co-localization-co-occurrence versus correlation. Journal of cell science. 2018 Feb 1;131(3).

3) Additional information about the genome-edited lines of *SABRE* is required, as beyond the genotyping data, no other confirmation evidence is provided. For instance, is it possible to check the gene expression at mRNA level or protein level? It was unclear to the reviewers if a complementation experiment was done to test the C-terminal tagged reporter lines. If it was, please alter the text to make this clearer. If not, what other evidence can the authors provide to make the case that *SABRE-3mNG* reflects endogenous localization?

4) Because the results here differ from previous work (e.g. Pietra et al), some discussion about how much the difference may reflect different cell types and organisms, rather than fundamental *SABRE* activities should be added. Specific cases to address are noted in the comments from the reviewers, below.

Reviewer #1 (Recommendations for the authors (required)):

This manuscript does a good job of explaining why the function of *SABRE* is worth pursuing, and it does reveal an unexpected behavior of the ER during cell division. With the caveat that this was studied in a different plant, it also brings into question previous ideas that *SABRE* is a MT regulator. But upon considering the manuscript as a whole, I did feel that there wasn't quite as much of an advance in our understanding of the molecular mechanisms of *PpSABRE* action as the Abstract lead me to expect.

One solution to this is some more examination of one clear phenotype. For example, the observation that *sabre* mutants have unusual nuclear migration patterns lead to the hypothesis that "*SABRE* may provide a link between the nuclear envelope and motors responsible for nuclear migration (Leong et al., 2020; Yamada and Goshima, 2018)." given the existence of reagents to monitor and inactivate these motors, it seems straightforward to test this model. Because the other functions of *SABRE* are more implied by localization than shown outright, this would be an experiment that would satisfy my concern about defining function.

Reviewer #2 (Recommendations for the authors (required)):

I am a bit uneasy about the sentence indicating "strong spatial correlation" of the ER and *SABRE* (specifically the word “strong”, and the implications). All that is said is "Pearson's correlation coefficients were calculated in NIS elements." I'd appreciate a few more details, since the finding is in contrast to the previous finding in Arabidopsis (Pietra et al., 2013) and image correlations can be fraught (e.g., Aaron et al., 2018). The temporal movement with the ER is convincing, but a few more details are warranted regarding the correlation, and I advise removing the subjective word "strong" as it appears to me that the puncta are associated with the ER (and *SABRE* is not an ER-resident protein).

Much of the data here is in contrast to Pietra et al., 2013, and it is not clear (to me, at least) if this is due to differences in cell types and possible multiple functions, vs. potential issues with interpretation of results. More experiments are not warranted, but a more thorough explanation in the Discussion is warranted. Things in particular I wonder about are:

– PPB alignment was observed to be misoriented in Pietra et al. – was this examined at all in this study?

– I am puzzled by the differences in cortical MT alignment found in their study, but not here. Could that be due to differences in cell developmental state (as cortical microtubules realign after cell expansion is complete in hypocotyl cells… I assume this is also true in *P. patens*?). Or perhaps, is there a feedback in Arabidopsis that influences the microtubules, i.e., is the a potential for an indirect effect? Or, is it possible that *SABRE* has different roles in different cellular contexts?

– Pietra et al., 2013, observed a cell plate enrichment, but no ER association – I think this should be discussed more thoroughly and perhaps speculated upon.

Aaron JS, Taylor AB, Chew TL. Image co-localization-co-occurrence versus correlation. Journal of cell science. 2018 Feb 1;131(3).

Reviewer #3 (Recommendations for the authors (required)):

In the manuscript titled “*SABRE* populates ER domains essential for cell plate maturation and cell expansion influencing cell and tissue patterning”, Cheng and Bezanilla characterized the function of the gene *SABRE* in the moss *Physcomitrium patens*, using molecular biology and cell biology approaches. The authors generated *sabre* mutant and overexpression lines using CRISPR-Cas9 mediated genome editing. Their detailed phenotypical analysis on the mutant and time-lapse data on *SABRE-mEGFP/3mNG* indicate that *SABRE* plays role in polarized cell growth, diffuse cell expansion and cell division. *SABRE* protein overlaps with the ER marker KDEL but not with the actin marker or microtubule marker. In *sabre* mutant, ER morphology becomes abnormal, and the process of cytokinesis is often defective. The authors thus demonstrated that *SABRE* regulates plant cell expansion and division via its interaction with the ER. These preliminary findings are potentially interesting, but are mostly descriptive and do not provide enough mechanistic understanding of the function of *SABRE* in ER organization and thus in cell morphogenesis and cell division. It is unclear whether the observed ER organization changes in *sabre* mutants are direct or indirect and how changes in ER organization in the mutant result in the cell growth defects and cell division defects.

1) The authors generated several genome-edited lines of *SABRE*, but other than genotyping data, no other confirmation evidence is provided. For instance, Are the mutants real knock-out? Is it possible to check the gene expression at mRNA level or protein level? For the C-terminal tagged lines, it is risky to conclude the fusion protein is functional just based on the “no growth defects in tagged plants”. Does the GFP signal reflect the real functional *SABRE*? I understand a rescue/complementation experiment here may be difficult, but a western blot showing fused *SABRE-mEGFP/3mNG* in the plants will help justifying the cell biology data shown later on.

2) I appreciate that the authors examined the phenotype of *sabre* mutants very carefully. It is clear that cell sizes and cell growth rates are both reduced, but based on the images in Figure 2D, there are more cells in the mutants, which is contradictory to the cell division failure described later. Could the authors quantify the cell number in the mutant vs. WT?

3) The authors believe *SABRE* is localized on the ER. Since *SABRE-3mNG* are detected on punctae, whereas ER-mCherry is shown as tubules and sheets, morphologically they are different. I feel at least one more piece of evidence (biochemical or cell biological evidence) is required here to support the conclusion.

4) The authors examined the cytoskeleton in *sabre* mutants based on fluorescent signals of two marker proteins. Both actin and microtubule still show accumulation in tips, how about the dynamics of these tips during cell expansion? The authors used “the correlation coefficient between images over all possible temporal spacings” to show actin/microtubule dynamics. Please explain more about the quantification data in Figure 2—figure supplement 1C and D in the main text.

5) The authors claim that actin plays a role in *SABRE* function. However, this is just based on the minor changes of actin marker protein in *sabre* mutant. Are the changes direct or indirect? What will happen to *SABRE* protein if actin is disrupted?

---

## [Author Response]

Essential revisions:1) The work here describes new localization and roles for a plant SABRE protein, and attempts to define the cellular mechanisms that underlie the defects observed upon loss of the protein in Physcomitrium. While it was appreciated that the authors touched on many different aspects of potential SABRE functions, the result is a somewhat diffuse manuscript, and the expectation that you would make sense of the phenotypes by explaining molecular functions of the SABRE protein. There are several ways to address this. One solution would be to investigate one hypothesis for SABRE function more in depth. For example, reviewer 1 suggests testing whether the author's hypothesis that SABRE may be the link to motors for nuclear migration. The other acceptable route would be to modify the text to highlight the findings in a way that people would appreciate the careful and comprehensive cellular analysis rather than expecting the molecular mechanism of SABRE.

As suggested, we revised the text emphasizing our cellular analysis and the novel localization.

2) The colocalization of SABRE with the ER are based on Pearson correlation coefficients (relative to other compartments) but details of the calculation are missing from the Materials and methods. Because localization is a major finding of the paper, and is in contrast to previous work, additional details on how this was measured should be included. Specific places to revise include: (1) “the correlation coefficient between images over all possible temporal spacings” to show actin/microtubule dynamics. Please explain more about the quantification data in Figure 2—figure supplement 1C and D in the main text. (2) The authors conclude that SABRE is ER-localized, but since SABRE-3mNG is detected in punctae, whereas ER-mCherry is shown as tubules and sheets, morphologically they are different. More evidence (biochemical or cell biological evidence) is required here to support the conclusion of ER-localization. In general, the authors should make sure their colocalization claims here are supported, as historically, there may have been over-confidence in imaging-based analyses.(see Aaron JS, Taylor AB, Chew TL. Image co-localization-co-occurrence versus correlation. Journal of cell science. 2018 Feb 1;131(3).

The correlation coefficient analysis used to analyze the dynamics of actin and microtubules at the cell cortex in Figure 2—figure supplement 1C and D was developed in 2010 (Vidali et al., 2010) and has been used by our lab and others (Arieti and Staiger, 2020; Burkart et al., 2015; de Bang et al., 2020; Gavrin et al., 2020; van Gisbergen et al., 2020, 2018) to describe dynamics of a variety of fluorescently labeled proteins. We now include a brief description in the revised manuscript in the Results and Materials and methods sections.

With respect to the association of *SABRE* puncta with ER structures, in the revised manuscript we developed an additional method to quantify the degree of overlap, which is outlined in Figure 4—figure supplement 3 and described in the Results and Materials and methods sections. This new method calculates the area of overlap between *SABRE* and the ER, actin, or microtubules. Unlike Pearson’s correlation, this method does not depend on fluorescence intensity and thus does not significantly change when *SABRE* is overexpressed, thereby providing a more robust measure of correlation between geometrically distinct structures. This new method is reported in the revised Figure 4 and the Pearson’s correlation coefficient was moved to Figure 4—figure supplement 3, as a second independent method to quantify the association of *SABRE* with the ER. We also provide more examples in Figure 4—figure supplement 3 of *SABRE* puncta moving along ER tubules demonstrating the sustained and dynamic interaction between *SABRE* and the ER. These new analyses and additional dynamic data, together with the fact that loss of *SABRE* function impacts the ER, suggests that *SABRE* associates with the ER and influences ER function.

3) Additional information about the genome-edited lines of SABRE is required, as beyond the genotyping data, no other confirmation evidence is provided. For instance, is it possible to check the gene expression at mRNA level or protein level? It was unclear to the reviewers if a complementation experiment was done to test the C-terminal tagged reporter lines. If it was, please alter the text to make this clearer. If not, what other evidence can the authors provide to make the case that SABRE-3mNG reflects endogenous localization?

We isolated the *SABRE* cDNA from wild type and *∆sab*. We cloned and sequenced the 5’ end of the cDNA and demonstrated that the cDNA from *∆sab* contains the predicted in-frame stop codons resulting from the edited allele. This is reported in Figure 1—figure supplement 1. We also used CRISPR-Cas9 coupled with homology directed repair to insert the stop cassette into the *SABRE* locus in the *OE-SAB-3mNG/ER-mCherry* line, thereby knocking out the overexpressed *SABRE-3XmNG*. We then imaged the *SAB-3mNG/ER-mCherry* parental line and the disrupted line (*∆sabre/OE-SAB-3mNG/ER-mCherry*). In Figure 4—figure supplement 2, we show that *∆sabre/OE-SAB-3mNG/ER-mCherry* no longer exhibits mNG fluorescence, in contrast to the robust signal in *OE-SAB-3mNG/ER-mCherry*.

In Figure 4—figure supplement 2 we show quantitative growth assays for the lines that were generated in this study. All the *SABRE* loci that were edited to generate a fluorescent protein fusion of SABRE grow indistinguishably from wild type. These experiments are, in essence, complementation experiments, because the only copy of *SABRE* present in these lines was fused to the coding sequence of a fluorescent protein. Given that disruption of *SABRE* function results in a strong growth defect and that lines that express only the fluorescent fusion protein from the endogenous *SABRE* locus have no growth defect and are indistinguishable from wild type, we reason that fluorescent fusions of SABRE are functional and thus represent the endogenous localization.

4) Because the results here differ from previous work (e.g. Pietra et al.), some discussion about how much the difference may reflect different cell types and organisms, rather than fundamental SABRE activities should be added. Specific cases to address are noted in the comments from the reviewers, below.

Discussion was added as outlined in specific responses below.

Reviewer #1 (Recommendations for the authors (required)):This manuscript does a good job of explaining why the function of SABRE is worth pursuing, and it does reveal an unexpected behavior of the ER during cell division. With the caveat that this was studied in a different plant, it also brings into question previous ideas that SABRE is a MT regulator. But upon considering the manuscript as a whole, I did feel that there wasn't quite as much of an advance in our understanding of the molecular mechanisms of PpSABRE action as the Abstract lead me to expect.One solution to this is some more examination of one clear phenotype. For example, the observation that sabre mutants have unusual nuclear migration patterns lead to the hypothesis that "SABRE may provide a link between the nuclear envelope and motors responsible for nuclear migration (Leong et al., 2020; Yamada and Goshima, 2018)." given the existence of reagents to monitor and inactivate these motors, it seems straightforward to test this model. Because the other functions of SABRE are more implied by localization than shown outright, this would be an experiment that would satisfy my concern about defining function.

We appreciate the reviewer’s concern and agree that the drafting of the initial manuscript was overstated. As suggested, we revised the text emphasizing our cellular analysis and the novel localization.

Reviewer #2 (Recommendations for the authors (required)):I am a bit uneasy about the sentence indicating "strong spatial correlation" of the ER and SABRE (specifically the word “strong”, and the implications). All that is said is "Pearson's correlation coefficients were calculated in NIS elements." I'd appreciate a few more details, since the finding is in contrast to the previous finding in Arabidopsis (Pietra et al., 2013) and image correlations can be fraught (e.g., Aaron et al., 2018). The temporal movement with the ER is convincing, but a few more details are warranted regarding the correlation, and I advise removing the subjective word "strong" as it appears to me that the puncta are associated with the ER (and SABRE is not an ER-resident protein).

We removed the word “strong”. Additionally, based on concerns due to relying solely on Pearson’s correlation coefficient to measure the co-localization of *SABRE* with the ER, we developed an independent method to quantify the degree of overlap between *SABRE* puncta and ER. For details, please see our response above to the essential revisions.

Much of the data here is in contrast to Pietra et al., 2013, and it is not clear (to me, at least) if this is due to differences in cell types and possible multiple functions, vs. potential issues with interpretation of results. More experiments are not warranted, but a more thorough explanation in the Discussion is warranted. Things in particular I wonder about are:– PPB alignment was observed to be misoriented in Pietra et al. – was this examined at all in this study?

*P. patens* protonemal cells do not have microtubule-based PPBs. It is still a debate in the field as to whether all cells in the gametophore have microtubule-based PPBs. Unfortunately, our microtubule reporter does not express well in gametophores so we did not explore this.

– I am puzzled by the differences in cortical MT alignment found in their study, but not here. Could that be due to differences in cell developmental state (as cortical microtubules realign after cell expansion is complete in hypocotyl cells… I assume this is also true in P. patens?). Or perhaps, is there a feedback in Arabidopsis that influences the microtubules, i.e., is the a potential for an indirect effect? Or, is it possible that SABRE has different roles in different cellular contexts?

We focused our analysis on the tip-growing protonemal cells that do not have the same cortical microtubule array as observed in Arabidopsis. We mention in the revised Discussion that these discrepancies may result from cell type differences.

– Pietra et al., 2013, observed a cell plate enrichment, but no ER association – I think this should be discussed more thoroughly and perhaps speculated upon.Aaron JS, Taylor AB, Chew TL. Image co-localization-co-occurrence versus correlation. Journal of cell science. 2018 Feb 1;131(3).et al.

We have expanded upon this in the revised Discussion. Unfortunately, Pietra et al. did not investigate *SABRE* and ER localization simultaneously during cell division, so it is difficult to definitively state that these results are at odds. In fact, we think that with VAEM imaging in Arabidopsis similar results may be obtained. Of note, the *SABRE* homolog in *Drosophila* localizes to the ER, suggesting that this association may be conserved.

Reviewer #3 (Recommendations for the authors (required)):In the manuscript titled “SABRE populates ER domains essential for cell plate maturation and cell expansion influencing cell and tissue patterning”, Cheng and Bezanilla characterized the function of the gene SABRE in the moss Physcomitrium patens, using molecular biology and cell biology approaches. The authors generated sabre mutant and overexpression lines using CRISPR-Cas9 mediated genome editing. Their detailed phenotypical analysis on the mutant and time-lapse data on SABRE-mEGFP/3mNG indicate that SABRE plays role in polarized cell growth, diffuse cell expansion and cell division. SABRE protein overlaps with the ER marker KDEL but not with the actin marker or microtubule marker. In sabre mutant, ER morphology becomes abnormal, and the process of cytokinesis is often defective. The authors thus demonstrated that SABRE regulates plant cell expansion and division via its interaction with the ER. These preliminary findings are potentially interesting, but are mostly descriptive and do not provide enough mechanistic understanding of the function of SABRE in ER organization and thus in cell morphogenesis and cell division. It is unclear whether the observed ER organization changes in sabre mutants are direct or indirect and how changes in ER organization in the mutant result in the cell growth defects and cell division defects.1) The authors generated several genome-edited lines of SABRE, but other than genotyping data, no other confirmation evidence is provided. For instance, Are the mutants real knock-out? Is it possible to check the gene expression at mRNA level or protein level? For the C-terminal tagged lines, it is risky to conclude the fusion protein is functional just based on the “no growth defects in tagged plants”. Does the GFP signal reflect the real functional SABRE? I understand a rescue/complementation experiment here may be difficult, but a western blot showing fused SABRE-mEGFP/3mNG in the plants will help justifying the cell biology data shown later on.

Please see our response above to the essential revisions.

2) I appreciate that the authors examined the phenotype of sabre mutants very carefully. It is clear that cell sizes and cell growth rates are both reduced, but based on the images in Figure 2D, there are more cells in the mutants, which is contradictory to the cell division failure described later. Could the authors quantify the cell number in the mutant vs. WT?

Interestingly, we did not observe the dramatic cell division failures in gametophores, suggesting that cell division may not be as affected in this tissue. We hope to pursue these differences in follow up studies.

3) The authors believe SABRE is localized on the ER. Since SABRE-3mNG are detected on punctae, whereas ER-mCherry is shown as tubules and sheets, morphologically they are different. I feel at least one more piece of evidence (biochemical or cell biological evidence) is required here to support the conclusion.

Given the very low level of *SABRE* protein, a biochemical approach would be extremely challenging. Thus, we have provided more examples of *SABRE* movement along ER tubules in Figure 4—figure supplement 3A, B. And as described above in the essential revisions, we performed a new analysis to quantify the area overlap between ER and *SABRE*. Please see our above response for additional details.

4) The authors examined the cytoskeleton in sabre mutants based on fluorescent signals of two marker proteins. Both actin and microtubule still show accumulation in tips, how about the dynamics of these tips during cell expansion? The authors used “the correlation coefficient between images over all possible temporal spacings” to show actin/microtubule dynamics. Please explain more about the quantification data in Figure 2—figure supplement 1C and D in the main text.

We have provided more information regarding the correlation coefficient analysis used to measure global actin and microtubule dynamics in the Results and Materials and methods sections. During periods of growth, actin and microtubules behaved similarly as shown in the revised Figure 2—figure supplement 1A, B where we present time projection images. The time projections are maximum intensity projections of frames from time-lapse acquisitions of growing cells. These projections illustrate the dynamic behavior of the microtubule and actin foci near the tip of growing cells. We were unable to identify any quantitative differences between wild type and *∆sabre* cells.

5) The authors claim that actin plays a role in SABRE function. However, this is just based on the minor changes of actin marker protein in sabre mutant. Are the changes direct or indirect? What will happen to SABRE protein if actin is disrupted?

We believe any impact on actin is likely indirect as disruption of actin does not affect *SABRE* localization and thus have removed statements implicating that *SABRE* affects actin.